# The Holocene sedimentary record of cyanobacterial glycolipids in the Baltic Sea: Evaluation of their application as tracers of past nitrogen fixation

Martina Sollai[1], Ellen C. Hopmans[1], Nicole J. Bale[1], Anchelique Mets[1], Lisa Warden[1], Matthias Moros[2] and Jaap S. Sinninghe Damsté[1,3]

[1]NIOZ Royal Netherlands Institute for Sea Research, Department of Marine Microbiology and Biogeochemistry, and Utrecht University, P.O. Box 59, 179AB Den Burg, Texel, The Netherlands.
[2]Leibniz Institute for Baltic Sea Research (IOW), Department of Marine Geology, Warnemünde, Germany.
[3]University of Utrecht, Faculty of Geosciences, Department of Earth Sciences, P.O. Box 80.021, 3508 TA Utrecht, The Netherlands.

*Correspondence to*: Jaap Sinninghe Damsté (*jaap.damste@nioz.nl*)

**Abstract.** Heterocyst glycolipids (HGs) are lipids exclusively produced by heterocystous dinitrogen-fixing cyanobacteria. The Baltic Sea is an ideal environment to study the distribution of HGs and test their potential as biomarkers because of its recurring summer phytoplankton blooms, dominated by a few heterocystous cyanobacterial species of the genera *Nodularia* and *Aphanizomenon*. A multicore and a gravity core from the Gotland basin were analyzed to determine the abundance and distribution of a suite of selected HGs at high resolution to investigate the changes in past cyanobacterial communities during the Holocene. The HG distribution of the sediments deposited during the Modern Warm Period (MoWP) was compared with those of cultivated heterocystous cyanobacteria, including those isolated from Baltic Sea waters, revealing high similarity. However, the abundance of HGs dropped substantially with depth and this may be caused by either a decrease in the occurrence of the cyanobacterial blooms or diagenesis, resulting in partial destruction of the HGs. The record also shows that the HG distribution has remained stable since the Baltic turned into a brackish semi-enclosed basin ~7200 Cal. yrs BP. This suggests that the heterocystous cyanobacterial species composition remained relatively stable as well. During the earlier freshwater phase of the Baltic (i.e. the Ancylus Lake and Yoldia Sea phases) the distribution of the HGs varied much more than in the subsequent brackish phase and the absolute abundance of HGs was much lower than during the brackish phase. This suggests that the cyanobacterial community adjusted to the different environmental conditions in the basin. Our results confirm the

potential of HGs as specific biomarker of heterocystous cyanobacteria in paleo-environmental
studies.
**1 Introduction**
Cyanobacteria are a broad and diverse group of photoautotrophic bacteria; they are found in
many terrestrial and aquatic environments (Whitton and Potts, 2012). They can exist as
benthos or plankton, unicellular or filamentous with or without branches, free-living or
endosymbionts (Rippka et al., 1979) and are of biogeochemical significance due to their role
in the cycling of carbon and nitrogen through photosynthesis and the fixation of $N_2$. However,
some $N_2$-fixing cyanobacteria can negatively impact aquatic ecosystems due to their role in
harmful algal blooms (HABs): exceptional events of phytoplankton growth causing
anomalous feedbacks on food webs, alteration in the geochemical features of the water
column (e.g. anoxia), and sometimes the release of harmful toxins in the environment.
Cyanobacterial HABs (cHABs) affect the surface of lacustrine, estuarine and tropical marine
environments worldwide; human-induced global warming and nutrient overload are blamed
for exacerbating the phenomenon (Paerl, 1988; Paerl et al., 2011; Paerl and Huisman, 2009).
The two processes of photosynthesis and $N_2$ fixation are theoretically incompatible
since the nitrogenase enzyme that catalyzes nitrogen fixation is inactivated by $O_2$. To cope
with this, $N_2$-fixing cyanobacteria have developed several strategies (Stal, 2009). The
filamentous diazotrophs of the orders *Nostocales* and *Stigonematales* spatially separate the
two metabolisms by forming special cells dedicated to the fixation of $N_2$, called heterocysts
(Wolk, 1982; Adams, 2000). Gas exchange is believed to be regulated by the heterocyst cell
wall, which consists of two separate polysaccharide and glycolipid layers (Murry and Wolk,
1989; Walsby, 1985) of which the latter acts as the gas diffusion barrier. These so called
heterocyst glycolipids (HGs) have been found to date to be unique to heterocyst-forming
cyanobacteria (Bryce et al., 1972; Nichols and Wood, 1968) and furthermore their
composition has been discovered to be distinct at the level of families and even genera
(Bauersachs et al., 2009a, 2014a; Gambacorta et al., 1998; Schouten et al., 2013). Their
structure comprises a sugar moiety glycosidically bound to a long *n*-alkyl chain (Fig. 1) with
an even number of carbon atoms (26 to 32) with various functional groups (hydroxyl and keto
groups) located at the C-3, $\omega$-1 and $\omega$-3 positions (Gambacorta et al., 1995, 1998; Schouten et
al., 2013). The sugar moiety of HGs found in non-symbiotic cyanobacteria is typically a
hexose (hereafter $C_6$) (Bryce et al., 1972; Lambein and Wolk, 1973; Nichols and Wood,

1968), while HGs associated with endosymbiotic heterocystous cyanobacteria have a pentose moiety (hereafter $C_5$) (Bale et al., 2015, 2017; Schouten et al., 2013). High-performance liquid chromatography coupled to electrospray ionization tandem mass spectrometry (HPLC/ESI-MS[2]) has emerged as a rapid method to analyze HGs in cultures (Bauersachs et al., 2009c, 2009a, 2014a) and modern day ecosystems such as microbial mats, lakes and marine systems (Bale et al., 2015, 2016, 2017; Bauersachs et al., 2009c, 2011, 2013, 2015; Wörmer et al., 2012).

$C_6$ HGs have been applied as specific paleo-biomarkers for the presence of $N_2$-fixing cyanobacteria in marine geological records back to the Pleistocene, and lacustrine deposits back to the Eocene and hence have provided evidence of the high potential for HGs preservation in sedimentary records (Bauersachs et al., 2010). In addition, temperature-induced modifications of the HG composition of heterocystous cyanobacteria were observed both in culture and in the environment and quantified by specific indices, suggesting the possible employment of HGs in reconstructing surface water temperatures (SWT) (Bauersachs et al., 2009a, 2014b, 2015). However, in general, the application of HGs as biomarker in environmental and paleo-environmental studies is still limited.

The Baltic Sea, characterized by the seasonal occurrence of cHABs mainly consisting of the HG-producing family *Nostocaceae*, presents an interesting location to both apply HGs as biomarkers in the present day system and to investigate their potential as proxies for reconstruction of past depositional environments. The modern Baltic, one of world's largest brackish bodies of water, is a shallow, semi-enclosed basin, characterized by estuarine circulation, having its only connection to the North Sea through the Danish straits (Fig. 2). Irregular winter inflows of marine oxygen-rich water, known as salinity pulses, represent the main mechanism of renewing and mixing of the bottom water, which otherwise experiences stagnation and increasing oxygen depletion with permanent stratification and persisting anoxia in its deep waters (Kononen et al., 1996). Since the last deglaciation (ca. 13-9 cal. kyr BP) the Baltic Sea has experienced specific hydrographical phases (Andrén et al., 2011). Following the ice retreat, the Baltic Ice Lake developed, which was followed by the Yoldia Sea phase, a short period when there was a connection with the sea. The subsequent Ancylus Lake phase (ca. 9.5–8.0 cal. kyr BP) was the last extended freshwater phase in the basin before a stable connection to the North Sea was established (Björck, 1995; Jensen et al., 1999). The transition phase began (ca. 7.8–7.3 cal. kyr BP) by a series of weak inflows of saline water, which eventually lead to the fully brackish Littorina Sea phase (~7.2–3.5 cal. kyr BP). The less brackish post-Littorina Sea phase (until ~1.3 cal. kyr BP) followed, and the

modern Baltic Sea is considered its natural continuation. In the last 1000 yr three alternating periods occurred: the Medieval Warm Period (MWP), the Little Ice Age (LIA) and the current Modern Warm Period (MoWP, starting at ~1950) (Kabel et al., 2012).

The modern Baltic undergoes summer cHABs primarily composed of a few species of filamentous heterocystous cyanobacteria *Nodularia spumigena*, *Aphanizomenon flos-aquae* and *Anabaena* spp. (Celepli et al., 2017; Hajdu et al., 2007; Hällfors, 2004; Kanoshina et al., 2003; Karjalainen et al., 2007; Ploug, 2008; Sivonen et al., 2007). Deep water anoxia, high phosphorus availability, calm water conditions and high irradiation resulting in relatively high sea surface temperature (SST) have been identified as main triggers for these blooms. Anoxic sediments lead to the release of phosphate in the water column, stimulating new cHABs and further enhancing anoxia, resulting in a reinforcing feedback (Finni et al., 2001; Kabel et al., 2012; Paerl, 2008; Paerl et al., 2011; Poutanen and Nikkilä, 2001; Stipa, 2002). The summer cHABs have been documented since the 19[th] century, with a reported increase in their frequency and intensity in the last 60 years, which has been related to human-induced eutrophication (Bianchi et al., 2000; Finni et al., 2001).

Several studies, based on fossil pigment and other paleo proxy records, suggest that cHABs have been recurring through the entire Holocene simultaneously with anoxic events and thus should be considered a natural feature of the basin, rather than a consequence of human impact (Bianchi et al., 2000; Borgendahl and Westman, 2007; Funkey et al., 2014; Poutanen and Nikkilä, 2001). SST has been suggested to have played an important role in these events (Kabel et al., 2012; Warden, 2017). Likely, at times of water column stratification and anoxia, high SST would have initiated cHABs in the basin, when exceeding a threshold temperature of ~16 °C, which is considered a trigger to the onset of the cHABs in the modern Baltic (Kononen, 1992; Wasmund, 1997). In addition, this would have enhanced the oxygen consumption of the deep water (Kabel et al., 2012).

The intrinsic occurrence of cHABs and their role in intensifying chronic anoxic events is not limited to the Baltic Sea. These same features have been observed in various stratified fresh water lakes in the Northern hemisphere (Fritz, 1989; McGowan et al., 1999; Schweger and Hickman, 1989; Züllig, 1986). However, there is no full agreement on this interpretation, as other authors argue that human perturbation has to be considered to be the main driving force behind the co-occurrence of cHABs with anoxia in the Baltic (Zillén and Conley, 2010). Therefore, more research is required to elucidate the relationship between recurring anoxic events and cHABs in the Baltic Sea.

In this study, we test the potential of HGs as paleo-proxy to investigate the changes in past communities involved in the summer cHABs in the Baltic Sea over the Holocene and the potential relationship with the anoxic events that occurred in the basin. To this end, a multicore and a gravity core from the Gotland basin were analyzed for HGs at high resolution. The results of the analysis were compared with the total organic carbon content and the nitrogen isotope record. This may help in further confirming the potential of HGs as specific biomarkers of heterocystous cyanobacteria in environmental studies.

## 2 Materials and methods

### 2.1 Sample site and sediment cores

Our sampling site is located in the Eastern Gotland Basin, one of the deepest basins (max. 248 m) within the Baltic Proper (Fig. 2). The gravity core (GC) 303600 (length 377 cm) was collected in the Gotland Basin (56˚55.02 N, 19˚19.98 W) at 175 m water depth during a cruise onboard the *R/V "Prof. Albrecht Penck"* in July 2009. The multicore (MUC) P435-1-4 (length 51.5 cm) was also collected in the Gotland Basin (56˚57.94 N, 19˚22.21 E) at 178 m water depth during cruise P435 onboard the *R/V "Poseidon"* in June 2012. The dating of the MUC and the brackish section of the GC was based on an age model, obtained by high resolution $^{14}$C dating of benthic foraminifera (Warden, 2017; Warden et al., 2017), which allowed us to date the MUC (as calculated kilo years before present, cal. kyr BP, and the corresponding AD) and the GC (as cal. kyr BP) back to 230 cm depth, which corresponds to ca. 7200 cal. yr BP.

The GC was cut in two halves and sub-sampled at high resolution with 1 cm slices from 0–237 cm and 2 cm slices from 237–377 cm. During the procedure depth 81–82 cm and 187–188 cm were missed. The MUC was sub-sampled at 0.5 cm resolution. The sediments obtained were freeze-dried and grounded before further analysis.

### 2.2 Elemental and stable isotope analysis

Sub-samples were taken from the GC sediment slices for determination of the total organic carbon (TOC) content at IOW and for the analysis of bulk stable nitrogen isotopes ($\delta^{15}$N) at NIOZ. The total carbon (TC) content of the sediments of the MUC and GC was measured by using an EA 1110CHN analyzer from CE Instruments, while a Multi EA- 2000 Elemental Analyzer (Analytic, Jena, DE) was employed to determine the total inorganic carbon (TIC). The TOC content was calculated as the difference between TC and TIC and expressed in

wt.%. The $\delta^{15}N$ was analyzed in duplicate on a Thermo Finnigan Delta Plus isotope ratio mass spectrometer (irmMS) connected to a Flash 2000 elemental analyzer (Thermo Fisher Scientific, Milan, Italy). Precision of the isotopes analysis was 0.2% for nitrogen measurements.

**2.3 Lipid extraction and analysis**

All slices from the MUC and alternating slices from the GC were extracted and analyzed for their HG content and distribution. Extraction was performed using an Accelerated Solvent Extractor (ASE 200, DIONEX; 100°C and $7.6 \times 10^6$ Pa) with a mixture of dichloromethane (DCM): methanol (MeOH) (9:1, v:v), to obtain a total lipid extract (TLE), which was dried under a flow of $N_2$. TLE was re-dissolved by sonication (10 min) in DCM/MeOH (1:1, v:v) and aliquots were taken and dried under a flow of $N_2$. These aliquots were dissolved in hexane, isopropanol and water (72:27:1, v:v:v) and filtered through a 0.45 μm regenerated cellulose syringe filter (4 mm diameter; Grace Alltech). Samples were analyzed by using a HPLC-triple quadrupole MS in multi-reaction monitoring (MRM) mode as described by Bale et al. (2015). For the analysis, an Agilent (Palo-Alto, CA, US) 1100 series HPLC with a thermostat-controlled auto-injector was employed coupled to a Thermo TSQ Quantum EM triple quadrupole MS equipped with an Ion Max source with ESI probe. The MRM method specifically targets $C_5$ and $C_6$ HGs with alkyl chains containing 26 and 28 carbon atoms (Bale et al., 2015). HGs were quantified as the integrated peak area per g of TOC (response units, r.u. gTOC$^{-1}$). The r.u. gTOC$^{-1}$ values were simplified for practical purpose by dividing them by $1 \times 10^{10}$. For the MUC, 30% of the samples were re-analyzed as duplicates; the calculated relative standard deviation was on average 5.3%. For all GC samples we performed the HPLC/MS$^2$ analysis twice; in this case the calculated relative standard deviation was on average 12.4%.

A selected number of samples was analyzed in full scan mode using an Ultra High Pressure Liquid Chromatography-High Resolution Mass Spectrometry (UHPLC-HRMS) method (Moore et al., 2013) as follows: we used an Ultimate 3000 RS UHPLC, equipped with thermostatted auto-injector and column oven, coupled to a Q Exactive Orbitrap MS with Ion Max source with heated electrospray ionization (HESI) probe (Thermo Fisher Scientific, Waltham, MA). Separation was achieved on an Acquity UPLC BEH HILIC column (150 x 2.0 mm, 2.1 μm particles, pore size 12 nm; Waters, Milford, MA) maintained at 30 °C. Elution was achieved with (A) hexane-propanol-formic acid-14.8 mol L$^{-1}$ aqueous NH$_3$ (79:20:0.12:0.04, v/v/v/v) and (B) propanol water-formic acid-14.8 mol L$^{-1}$ aqueous NH$_3$

(88:10:0.12:0.04, v/v/v/v) starting at 100% A, followed by a linear increase to 30% B at 20
min, followed by a 15 min hold, and a further increase to 60% B at 50 min. Flow rate was 0.2
ml min$^{-1}$, total run time was 70 min, followed by a 20 min re-equilibration period. Positive ion
ESI settings were: capillary temperature, 275°C; sheath gas (N$_2$) pressure, 35 arbitrary units
(AU); auxiliary gas (N$_2$) pressure, 10 AU; spray voltage, 4.0 kV; probe heater temperature,
275°C; S-lens 50 V. Target lipids were analyzed with a mass range of m/z 350–2000
(resolution 70,000 ppm at m/z 200), followed by data-dependent tandem MS$^2$ (resolution
17,500 ppm), in which the ten most abundant masses in the mass spectrum were fragmented
successively (normalized collision energy, 35; isolation width, 1.0 m/z). The Q Exactive was
calibrated within a mass accuracy range of 1 ppm using the Thermo Scientific Pierce LTQ
Velos ESI Positive Ion Calibration Solution. During analysis dynamic exclusion was used to
temporarily exclude masses (for 6 s) in order to allow selection of less abundant ions for MS2.
A number of indices have been suggested to express correlation between the
distribution of HGs and growth temperature (Bauersachs et al., 2009a, 2014b, 2015). We
examined our data using two such indices, the HDI$_{26}$ and the HDI$_{28}$ (heterocyst diol index of
26 and 28 carbon atoms, respectively), defined as follows:

$$\text{HDI}_{26} = \frac{\text{HG}_{26}\ \text{diol}}{\text{HG}_{26}\ \text{keto} - \text{ol}\ +\ \text{HG}_{26}\ \text{diol}} \quad (1)$$

$$\text{HDI}_{26} = 0.0224 \times SWT + 0.4381; r^2 \quad (2)$$
$$= 0.93$$

$$\text{HDI}_{28} = \frac{\text{HG}_{28}\ \text{diol}}{\text{HG}_{28}\ \text{keto} - \text{ol}\ +\ \text{HG}_{28}\ \text{diol}} \quad (3)$$

$$\text{HDI}_{28} = 0.0405 \times SWT + 0.0401; r^2 \quad (4)$$
$$= 0.70$$

SWT = surface water temperature. These SWT calibrations have been determined in a study
of a freshwater lake (Lake Schreventeich, Kiel, Germany; Bauersachs et al., 2015).
**2.4 Data analysis**
Principal component analysis (PCA) was performed with the R software package for
statistical computing, to test the variation observed in the HGs distribution.

## 3 Results

### 3.1 Sediment core characteristics

The basin has experienced periodical anoxic bottom waters, which resulted in the alternating deposition of laminated and homogeneous sediments (Fig. 3; see also Andrén et al. (2000)). The sediments of the MUC represent almost 1000 yr of sedimentation and comprise the MoWP (~0–11 cm depth, corresponding to ~2012–1950 AD or -0.06 to 0 cal kyr BP), the LIA (~12–41 cm, corresponding to ~1950–1260 AD or ~0.1–0.7 cal kyr BP) and almost the entire MWP (~42–52 cm, corresponding to ~0.7–0.9 cal kyr BP). The upper part of the GC overlaps with the deeper part of the MUC (i.e. ~0 to 17 cm depth in the GC roughly corresponds to ~35 to 52 cm of the MUC). The upper part of the GC covers the initial phases of the LIA (until ca. 6 cm, ~0.6 cal kyr BP), the complete Littorina Sea and Ancylus Lake stages, down to part of the Yoldia Sea stage.

### 3.2 Abundance and distribution of HGs

In total 104 sediment horizons of the MUC and 153 horizons of the GC were analyzed for $C_6$ and $C_5$ HGs with alkyl chains with 26 and 28 carbon atoms using HPLC-triple quadrupole MS in multi-reaction monitoring (MRM) mode as described by Bale et al. (2015). Bauersachs et al. (2017) have recently analyzed the HGs of eight representative heterocystous cyanobacterial strains isolated from the Baltic Sea and the six $C_6$ HGs targeted in our study form by far the majority (i.e. 97.7-100%) of the HGs of these strains. HGs with longer alkyl chains were not detected, suggesting that, at least for the brackish phase, our analysis method will provide a proper view of changes in the overall HG distribution.

$C_5$ HGs were not detected at all, but the targeted $C_6$ HGs were present in all samples of both cores. The $C_6$ HGs detected in this study were: 1-(O-hexose)-3,25-hexacosanediol ($C_{26}$ diol HG; see Fig. 1 for structures); 1-(O-hexose)-3-keto-25-hexacosanol ($C_{26}$ keto-ol HG); 1-(O-hexose)-3,27-octacosanediol ($C_{28}$ diol HG); 1-(O-hexose)-3-keto-27-octacosanol ($C_{28}$ keto-ol HG); 1-(O-hexose)-3,25,27-octacosanetriol ($C_{28}$ triol HG); 1-(O-hexose)-27-keto-3,25-octacosanediol ($C_{28}$ keto-diol HG). A selected number of samples from the brackish phase was also analyzed in full scan mode to check for the presence of HGs with longer alkyl side chains but these were not encountered (Table 2). The HG distribution obtained using this method was comparable to that obtained with the HPLC-triple quadrupole MS method.

The distribution of the six quantified HGs changed substantially with depth (Fig. 4). The $C_{26}$ diol HG was the dominant component, accounting for ~50 to 95% of the HGs in the

sediments recording the brackish phase of the basin. In the sediments deposited during the Ancylus Lake and Yoldia phase (i.e. below 213 cm of the GC) the fractional abundance of the $C_{26}$ diol HG was more variable, reaching only 20–30% at some discrete depths. In the sediments deposited during the brackish phase the fractional abundance of all keto HGs (i.e., $C_{26}$ keto-ol HG, $C_{28}$ keto-ol HG and $C_{28}$ keto-diol HG) diminished with increasing depth, roughly from 3–15% to <2% (Fig. 4b). In the sediments deposited during the Ancylus Lake and Yoldia Sea phase, however, their fractional abundance showed more variation and in general it increased and reached ~10–40% at some specific depths. The fractional abundance of the $C_{28}$ diol HG remained steady for most of the sediments deposited during the brackish phase (~10% on average), although slightly increased values occurred in the oldest part of the brackish section, up to ~15% (Fig. 4b). In the Ancylus Lake and Yoldia Sea section the fractional abundance of the $C_{28}$ diol HG was higher, with values sometimes reaching almost 60%, but also more variable. The fractional abundance of the $C_{28}$ triol HG was <2% for most of the sediments deposited during brackish phase, with the exceptions of the shallower (8–16%) and the deeper part, close to the boundary with the freshwater phase (3–9%). In the Ancylus Lake and Yoldia Sea sections the relative abundance of the $C_{28}$ triol HG generally remained <2%, although it was between 3–11% in several horizons in the deeper part (Fig. 4b). For the Ancylus Lake and Yoldia Sea section we did not check the general distribution of the HGs and, therefore, cannot exclude that HGs with alkyl chains >28 carbon atoms occur during these intervals.

The $C_6$ HG abundance (sum of the six $C_6$ HGs; hereafter referred to as HG abundance) profile showed four peaks in the first 8 cm of the MUC of respectively 144, 82, 117 and 69 r.u. gTOC$^{-1}$ (Fig. 3a). After this last peak, the abundance of the HGs decreased substantially by a factor ~30 in some cases (i.e., ~5 r.u. gTOC$^{-1}$) and remained at this level with increasing depth over the whole of the MUC (Fig. 3a).

The HG abundance in the upper part of the GC (up to ~11 cm) was 3 to 6 times higher (7 to 18 r.u. gTOC$^{-1}$) than that recorded in the corresponding fraction of the MUC (2 to 4 r.u. gTOC$^{-1}$). At ~17 cm of the GC, which is equivalent to ~52 cm or the bottom of the MUC, the abundance were in the same order of magnitude (4 to 5 r.u. gTOC$^{-1}$). Between ~25 and 213 cm depth (~1.3–7.1 cal kyr BP) the abundance of the HGs decreased substantially further by a factor of ca. 6 to 10, with the exception of several small peaks at discrete depths (respectively, ~5 r.u. gTOC$^{-1}$ at ~35 cm; ~4 r.u. gTOC$^{-1}$ at ~53 cm, at ~92 cm and at ~108 cm; ~3 r.u. gTOC$^{-1}$ at ~188 cm). Deeper in the core (213–375 cm; i.e. the Ancylus Lake and Yoldia phase) the abundance of the HGs was even lower (Fig. 3a).

**3.3 Principal component analysis of the HG distribution**
The variation observed in the HG distribution in the sediments was examined by applying a
principal component analysis (PCA) to the fractional abundances of the six HGs (Fig. 5). The
first two principal components (PCs) explained most of the variation observed, accounting for
47 and 29% of the variance, respectively (Fig. 5a). The first principal component (PC1)
showed a positive loading of all keto HGs and of the $C_{28}$ triol HG. Specifically, the $C_{26}$ keto-
ol HG and the $C_{28}$ keto-diol HG had the most positive loading (Fig. 5a). The $C_{26}$ diol HG was
the only component showing a negative loading in PC1; the $C_{28}$ diol HG did not show any
loading on PC1. PC2 is primarily determined by the positive loadings of the $C_{28}$ diol and
keto-ol HGs, whereas all other HGs had negative loadings on PC2.
Figure 5b shows the scores of all analyzed sediment horizons on PC1 and PC2, which
reveals clearly defined different signatures. The brackish phase sediments all scored
negatively or just above zero on PC2. However, the score on PC1 was more variable; the
MoWP sediments scored most positively on PC1, whereas the pre-MoWP brackish sediment
scored less positive on PC1, which is due to the higher fractional abundances of the $C_{26}$ keto-
ol and $C_{28}$ keto-diol HGs in the MoWP sediments. The remaining sediments of the Ancylus
Lake I and Yoldia Sea phase all scored positively on both PC1 and PC2 and therefore
distinctly from the brackish phase sediments but also showed much more variability. The
sediments of the Ancylus Lake transitional phase II (filled triangles in Fig. 5b) plotted much
closer to those of the brackish phase with some data points with similar PC1 and PC2 values.
Figure 4 shows the variation of the scores on PC1 and PC2 with depth. The sediments
of the MUC exhibited a decreasing trend in PC1 with increasing depth, caused by the
reduction in the fractional abundance of the positively scoring keto HGs, in favor of the
negatively scoring $C_{26}$ diol (Fig. 4a). For the GC (Fig 4b), the PC1 scores varied between -2
and -1, from the top up to 213 cm depth (i.e. the brackish phase), consistent with the
dominance of the $C_{26}$ diol HG in this section. At greater depth (i.e. the Ancylus Lake and
Yoldia Sea phases) large variations in the score of PC1 were observed (Fig. 4b). Scores were
mostly positive; negative PC1 scores were only found at three discrete depths, i.e. 239, 303
and 343 cm. The generally positive score in these phases highlights the greater contribution of
HGs other than $C_{26}$ diol HG. The PC2 score of the sediments of the MUC was constantly
around -1, (Fig. 4a). In the GC, PC2 was close to zero during the brackish water phase (Fig.
4b). In the sediments of the Ancylus Lake and Yoldia Sea phases the PC2 score was generally
positive, clearly influenced by the higher fractional abundance of positively scoring $C_{28}$ diol
and $C_{28}$ keto-ol HGs, but variable.
**4 Discussion**
This study investigates the presence of HGs in the recent sedimentary record of the Baltic Sea
and represents the first attempt to relate them with the recurring anoxic events that took place
in the basin during the Holocene as well as the ongoing increase in cHAB over the last 60
years. In our data set we recognized various phases, characterized by different distributions of
HGs (cf. Figs. 4 and 5b). Here these records and their implications for the heterocystous
cyanobacterial community composition are discussed.
**4.1 The distribution of HGs**
The composition of HGs in cyanobacteria is known to be related to their taxonomy
(Bauersachs et al., 2009a, 2014a, Gambacorta et al., 1995, 1998; Schouten et al., 2013;
Wörmer et al., 2012). Hence we compared the distribution of the HGs observed in our
sedimentary record of the Baltic Sea with the HGs produced *in vitro* by different
heterocystous cyanobacterial species.
**4.1.1 Brackish sediments**
Firstly, the most recent sediments (MoWP, <11 cm depth of MUC) were compared with
species that thrive in the modern Baltic Sea (Table 1). The recurring late summer (July-
August) cHABs of the Baltic are dominated by the taxa *Nodularia spumigena*,
*Aphanizomenon flos-aquae* and, to a minor extent, by *Anabaena* spp. and other species from
the order *Nostocales*, family *Nostocaceae* (Hajdu et al., 2007; Hällfors, 2004; Kanoshina et
al., 2003; Karjalainen et al., 2007; Sivonen et al., 2007; Celepli et al., 2017). While the
*Nodularia* genus is usually prevalent, changes in the composition of the community have been
observed from the early to the late stage of the cHAB and from one year to another, resulting
in a large variation of its features over time (Finni et al., 2001; Hajdu et al., 2007; Kahru et al.,
1994; Wasmund, 1997). A recent extensive meta-omics study revealed that in the Baltic
proper (the predominant area for cHABs) 69% of the heterocystous cyanobacteria belong
*Aphanizomenon*, 23% to *Anabaena*, and 8% to *Nodularia* (Celepli et al., 2017).
The HG distribution in the MoWP sediment, with the $C_{26}$ diol as the dominant HG
(Fig. 4a, summarized in Table 1), agrees well with the HG distribution in cultures of
*Nodularia, Aphanizomenon* and *Anabaena* as well as other members of the *Nostocaceae*

family (Table 1), including those that have been isolated from the Baltic (Bauersachs et al., 2009a, 2017). These cultures generally also synthesized minor amounts of the $C_{26}$ keto-ol HG, as was seen in the MoWP sediments. The $C_{28}$ diol, present in trace amounts in the MoWP sediments, was found in varying amounts in the *Nodularia*, *Aphanizomenon* and *Anabaena* cultures. Even between different strains of the same species, amounts present were highly variable from a dominant component to not detected (Table 1). The $C_{28}$ keto-ol, $C_{28}$ triol and $C_{28}$ keto-diol HGs were minor components in the MoWP sediment. While not produced consistently across the *Nodularia*, *Aphanizomenon* and *Anabaena* cultures, they were found in certain strains, generally as trace or minor components, in agreement with the distribution in the sediment (Table 1). It is possible, however, that the presence of the $C_{28}$ triol HG in the MoWP sediments may be linked to the presence of the genus *Calothrix* (cf. Table 1), which is commonly found in the rocky seabed of the basin (Sivonen et al., 2007).

Overall, the distribution of the HGs observed in the MoWP sediments was in good agreement with the HG distribution of the family *Nostocaceae* (Table 1), which fits with the reported dominance of members of this family during the summer cHABs of the Baltic. Furthermore, the HG distribution remained relatively constant throughout the MoWP sediments (Fig. 4a), suggesting that overall the community composition of heterocystous cyanobacteria in the Baltic Sea has remained stable during the last ~60 years.

The HG distribution in the sediment from the pre-MoWP brackish phase (i.e. from the Ancylus Lake-Littorina Sea (AL-LS) transition to the start of the MoWP) reconstructed in this study was similar to that of the MoWP, although the $C_{26}$ diol and the $C_{28}$ diol were present in a greater fractional abundance (Table 1; Fig. 4). The other four HGs were either minor or occurred in traces. Although often absent, a number of *Nostocaceae* strains have been found to contain the $C_{28}$ diol (Table 1), and in one *Anabaena* sp. strain (CCY9402) it was found to be the dominant HG (Bauersachs et al., 2009a). The increased proportion of the $C_{28}$ diol through the pre-MoWP brackish phase suggests there was a somewhat different cyanobacterial community composition than during the MoWP, although most probably still dominated by cyanobacteria belonging to the family *Nostocaceae*. The HG distribution remained relatively constant from the establishment of the brackish phase to the MoWP (Fig. 4), which suggests that the cyanobacterial community of the Baltic did not undergo major changes from the AL-LS transition to the MoWP and remained dominated by cyanobacteria belonging to the family *Nostocaceae*.

**4.1.2 The Ancylus Lake sediments**

The Ancylus Lake phase displayed a distinct HG distribution from the brackish phase (Fig. 4b; summarized in Table 1). The $C_{28}$ diol was often dominant and both the $C_{26}$ and $C_{28}$ keto-ol were present in a higher proportion than during the brackish phase. This is most evident for the Ancylus Lake phase I and the middle section (ca. 230-210 cm) of the Ancylus Lake phase II. Yet, at the first (ca. 250-230 cm) and last part (ca. 210-193 cm) of the Ancylus Lake transitional phase II, the HG distribution is more similar to the one observed in the brackish phase (Fig. 4b). This is also evident from the PCA analysis with more negative values for PC1 and PC2 at those depths (Figs. 4b and 5b). The AL-LS transition did not happen instantly (Borgendahl and Westman, 2007; Emeis et al., 1998; Gustafsson and Westman, 2002; Hyvarinen, 1984) and probably the sediment intervals showing a brackish-like distribution of the HGs correspond to weak pulses of marine water that might have occasionally entered the basin already during the Ancylus Lake transitional phase II and consequently influenced the overall distribution of the HGs (Fig. 4b). This final stage of this transition is also evident from the lithology and TOC profile (Fig. 3c).

When the Baltic evolved from a freshwater lake into a brackish semi-enclosed basin, it experienced an increase in salinity from fresh to values of 10–15 ‰ (Gustafsson and Westman, 2002). The observed changes in the HG distribution over the AL–LS transition suggest that this change from freshwater to brackish resulted in a different cyanobacterial species composition and hence a different HG distribution. Indeed, several freshwater species have been found to contain a HG distribution dominated by the $C_{28}$ diol (Table 1), including *Cyanospira rippkae* (Soriente et al., 1993), *Tolypothrix tenuis* (Gambacorta et al., 1998) and *Aphanizomenon aphanizomenoides* (Wörmer et al., 2012), although we emphasize that we did not analyze HGs with $C_{28+}$ alkyl chains for this stage and, therefore, cannot exclude the contributions of cyanobacteria producing such extended HGs. Alternatively, an increased influx of soil organic matter during the Ancylus Lake phase could be responsible for the distributional HG changes. However, since HG lipids contain an attached sugar moiety, we feel it is unlikely that HGs produced in soil will make it to the sediments of the Baltic Sea since they would be exposed extensively to oxygen during transport and only relatively stable components such as lignin, wax lipids, and branched GDGTs will likely survive this transport to the middle of the Baltic Sea where our core was taken.

For *Nodularia spumigena,* the most abundant heterocystous cyanobacterium in the present Baltic, its basic physiological features, such as growth, production of the toxin nodularin and differentiation of heterocysts are substantially affected at extreme salinities (Mazur-Marzec et al., 2005; Moisander et al., 2002). This is thought to be the predominant

reason why *Nodularia* blooms only occur within a certain salinity range (i.e. 7–18‰) in
nitrogen-deficient waters (Mazur-Marzec et al., 2005). This would imply that during the
Ancylus Lake phase the low salinity was limiting the growth of *Nodularia* sp.. Other
heterocystous cyanobacteria such as *Anabaena* and *Aphanizomenon* may be better adapted to
freshwater conditions.
**4.1.3 Yoldia Sea sediments**
Also for the Yoldia Sea sediments a high variability is observed in the HG distribution (Figs.
4b, 5b). The most distinct feature is the relatively high fractional abundance of the $C_{28}$ diol
HG, which reaches sometimes 50%, the highest value recorded for all sediments. The Yoldia
Sea phase was a relatively short period when a connection with the sea was established and
waters may have become brackish. Nevertheless, the HG distribution is not at all similar to
that of the brackish phase.
**4.1.4 Does the distribution of the fossil HGs records a paleotemperature signal?**
As a consequence of the retreat of the ice sheet and the inlet of the sea water through the
Danish straits, there was an increase of water temperature during the AL–LS transition
(Björck, 1995). It is possible that this increase in water temperature could have been
responsible for the changes in the HG distribution, as growth temperature has been reported to
affect the distribution of the HGs in cyanobacteria belonging to the order *Nostocales*
(Bauersachs et al., 2009a, 2014b, 2015). Specifically, increasing temperature positively
correlated with increasing relative proportions of HG diols over HG keto-ols. In our record,
the ratio of diols to keto-ols increased from the Ancylus Lake towards the brackish phase
(Fig. 4b), which would be in agreement with the higher SWTs during the brackish phase.
However, when the HG proxies are used to estimate SWT based on the proxy calibrations
from a lake (Eq. 1-4), the predicted temperatures are somewhat unrealistic. For the brackish
phase the $HDI_{26}$ and $HDI_{28}$ values vary between 0.96-1.00 and 0.95-1.00, translating in
average SWT of ca. 24 and 23°C, respectively. This is too high, even for summer
temperatures when the cHABs occur (Kanoshina et al., 2003). $TEX_{86}$-derived summer
temperatures (Kabel et al., 2012; Warden et al., 2017) do not exceed 17.5°C (Fig. 3d).
Application of the HG-based calibrations in this setting assumes that salinity has no impact
since they have been established for a freshwater lake (Bauersachs et al., 2015). For the
Ancylus Lake and Yoldia Sea phases the $HDI_{26}$ and $HDI_{28}$ values are highly variable and
range between 0.52-1.00 and 0.00-0.99, translating in average SWTs of ca. 20 and 17°C,

respectively. This is lower than observed for the brackish phase but also seems too high. Apparently, cyanobacterial species composition exerts an important control on the HG distribution in such a way that the HGs are not able to predict accurate temperatures in the brackish/freshwater system of the Baltic. Cultivation experiments with HG-producing strains isolated from the Baltic Sea (see Table 1) at varying temperatures may improve HG palaeothermometry of Baltic Sea sediments.

**4.2 The abundance of HGs**

**4.2.1 Is HG abundance a good measure for cHABs and anoxic events?**

In the Baltic the occurrence of summer cHABs has intensified since the 1950s (Kabel et al., 2012; Poutanen and Nikkilä, 2001). Yet, due to the spatial patchiness and inter-annual variability, it has proven difficult to recognize a clear trend of the cHABs at the scale of the entire Baltic (Finni et al., 2001; Kahru and Elmgren, 2014; Pitarch et al., 2016; Wasmund and Uhlig, 2003). However, the general interest towards these events has led to intensified research (see Finni et al., 2001; Kahru and Elmgren, 2014; Kutser et al., 2006 among others) and to the establishment of the Baltic Marine Environment Protection Commission (HELCOM) in 1992 to monitor this phenomenon. Disparate indices and parameters have been employed to describe and quantify cHABs over time, and were applied in the different areas of the Baltic, which are biogeochemically heterogeneous and display distinct seasonal dynamics (Kahru, 1997; Kahru et al., 2007; Kahru and Elmgren, 2014; Kononen, 1992; Kutser et al., 2006; Pitarch et al., 2016; Wasmund and Uhlig, 2003). The methods employed and the frequency of the sampling campaigns have improved in the recent past, reducing the inaccuracy associated to previous sampling methods and measurements (Hansson and Öberg, n.d.; Kahru, 1997; Kahru and Elmgren, 2014; Wasmund and Uhlig, 2003). However, intrinsic limitations of the techniques in use may still cause difficulties when comparing measurements from different years, even within the same time series (Finni et al., 2001; Kahru, 1997; Kahru and Elmgren, 2014).

Here, the HG abundance over the past ~30 years (i.e. 2012–1979 of the MoWP), recorded within the first ~7 cm of the MUC are discussed in comparison with a time series of the cHABs episodes relative to the Eastern Gotland Basin (Fig. 6), whose intensity is expressed as the frequency of cyanobacteria accumulation (FCA) (Kahru and Elmgren, 2014). FCA is determined by ocean color satellite data and expresses the frequency of the occurrence of cHABs in July-August using 1 $km^2$ pixels (Kahru et al., 2007). Kahru and Elmgren (2014) reported prominent cHABs in the early 1980s, in the period 1990–1996 and again from 1999

until 2008, with the interval 2005–2008 recording the highest FCA percentages, whilst with relevant inter-annual changes of the areal extent (Kahru, 1997; Kahru et al., 1994, 2007; Kahru and Elmgren, 2014). The HG lipid biomarker abundance profile from our sampling site was overall in reasonable agreement with the FCA measurements (Fig. 6). However, it failed to record the intense cHABs of the early 1980s, and there is a mismatch of one or two years in recording the start of the strong cHABs recorded at the end of the same decade (Kahru and Elmgren, 2014). Furthermore, this comparison is complicated by a certain degree of uncertainty in the age model of the sedimentary record. Moreover, the intrinsic temporal and spatial variability of the cHABs in the modern Baltic Sea, together with the difficulties encountered in the attempt of creating a consistent long time series that combines FCA data from multiple satellite sensors may provide an explanation for the discrepancies observed (Kahru and Elmgren, 2014; Wasmund and Uhlig, 2003).

We observed multiple peaks of the HGs absolute abundance in the MoWP section of the MUC core ($\leqslant$11 cm depth), which reached ~50–150 r.u. gTOC$^{-1}$. Below this in the LIA section, the HG abundance declined sharply to <10 r.u. gTOC$^{-1}$ (Fig. 3a). This decline may be expected given that the MoWP is characterized by higher summer surface temperature (Fig. 3d), increased organic matter deposition, and more frequent anoxic events than the LIA phase (Kabel et al., 2012), all conditions that lead to increased cHABs. Furthermore, the cooler LIA experienced more oxygenated bottom water, which may have affected HG preservation (see also below). However, a substantially increased HG abundance was not observed below the LIA in the MWP section of the MUC core (Fig. 3a). Similar to the MoWP period, the MWP was characterized by higher summer temperatures (Fig. 3d) and increased stratification of the water column that would favor bottom anoxia and, presumably, the occurrence of cHABs. The top of the GC also records the LIA–MWP transition (Fig. 3). Here, the HGs abundance reached ~10–18 r.u. gTOC$^{-1}$ at <30 cm depth, which is up to 4 times higher than the HGs abundance observed in the MUC for the same period. This discrepancy between the HGs records in the two related cores is puzzling. After the MWP, HG abundance declined to $\leqslant$5 r.u. gTOC$^{-1}$ during the remaining part of the brackish phase, as recorded in the GC (Fig. 3a), with only a minor increase of the HG concentration during the periods when summer temperature was higher and the Baltic Sea was stratified, resulting in bottom water anoxia (Fig. 3; e.g. during the Holocene thermal maximum).

Based on these data from the Baltic Sea, it is not possible to confidently couple the HG abundance record directly to cHAB occurrences and anoxic events in the past. Several

factors are thought to affect this relationship. Firstly, it is possible that the occurrence of cHABs varied over time. In the shallow part of both sediment cores, HGs absolute abundance was generally high, but it started declining with increasing depth, independently from other factors (Fig. 3). This might suggest that cHABs were less common and intense in the past brackish Baltic Sea, even at times of warmer and more stratified conditions. Secondly, the succession of oxic/anoxic bottom water conditions may impact the preservation efficiency of HGs. Such successions took place in the Baltic Sea during the entire Holocene as is evident from the alternation of dark–laminated with light–homogeneous sections in the sedimentary record (Kabel et al., 2012). In the shallow part of both sediment cores, the high absolute abundance of HGs coincided with dark–laminated sediment phases; low HGs on the contrary, concurred with light–homogeneous phases. In contrast, in the deeper part of the section this correspondence was lost. Finally, the generally declining trend of the HGs absolute abundance in the shallow sediments might also be due to anaerobic breakdown of the HGs. A decline of lipid biomarkers with depth has been documented before in anoxic Black Sea surface sediments (Sun and Wakeham, 1994). This process would be seemingly in contrast with previous indications of a high preservation potential of the HGs in ancient marine and lacustrine anoxic sediments (Bauersachs et al., 2010), but it should be realized that even in the older Baltic Sea sediments HGs are still detected. Apparently, even if diagenesis is occurring, it does not result in complete destruction of HGs.

The HG results seem to partly contrast an earlier study that, based on fossil pigment records, suggested that cHABs have been recurring simultaneously with the mid-Holocene anoxic events (Funkey et al., 2014). However, this study used carotenoids (i.e. zeaxanthin and echinenone) that are not entirely specific for cyanobacteria and are certainly not limited to nitrogen-fixing cyanobacteria, as opposed to the highly specific HGs that were used here. For example, zeaxanthin is also produced by *Synechococcus*, the dominating unicellular cyanobacterial species in the Baltic Sea (Celepli et al., 2017). Furthermore, in this environment of highly variable sediment redox conditions the effect of diagenesis should be considered. Carotenoids are amongst the most unstable organic biomarkers because of their very labile conjugated system of double bonds. Changes in redox conditions of bottom and sediment pore waters will thus have a major effect on the concentration of carotenoids and this may explain the enhanced concentration of carotenoids in the mid-Holocene TOC-enriched sections (Funkey et al., 2014).

**4.2.2 Changing abundance of the HGs over the AL-LS transition**

The general down-core decrease in the HGs abundance throughout the brackish phase is
continued into the Ancylus Lake and Yoldia Sea phases, where the HG abundance is at least
an order of magnitude lower that in the first part of the brackish phase (Fig. 3a). The lower
HG abundance in the Ancylus Lake and Yoldia Sea phases, relative to the brackish phase,
could indicate that $N_2$-fixing cyanobacteria were much less abundant during this freshwater
phase. Indeed, further evidence for a lower abundance of diazotrophic phytoplankton during
the Ancylus Lake and Yoldia Sea phases comes from the record of $\delta^{15}N$ values (Fig. 3b).
During these phases the $\delta^{15}N$ values are 4–6 ‰, indicating that most of the phytoplankton
community was relying on ammonium or nitrate as nitrogen sources rather than atmospheric
nitrogen (Bauersachs et al., 2009b; Emerson and Hedges, 2008). When other forms of
nitrogen are abundant the energetically expensive $N_2$ fixation becomes disadvantageous
(Arrigo, 2005; Capone et al., 2005; Karl et al., 1997). At the start of the LS phase, $\delta^{15}N$ values
drop to 1–3‰ (Fig. 3b), a range expected when $N_2$-fixing cyanobacteria contribute
substantially to primary production (Bauersachs et al., 2009b; Rejmánková et al., 2004;
Zakrisson et al., 2014), and remained in this range.
As discussed above, the salinity change from a freshwater lake to a brackish sea may
had a significant effect on the heterocystous cyanobacterial composition in the Baltic. This
environmental change may have also been a cause of the increased abundance of
heterocystous cyanobacteria. Another environmental factor change that could have promoted
increased blooming of heterocystous cyanobacteria is the increase in water temperature over
the AL–LS transition (Björck, 1995). Temperature is a crucial factor influencing the growth
rate and other metabolic features of free-living heterocystous cyanobacteria (Bauersachs et al.,
2014b; Kabel et al., 2012; Mazur-Marzec et al., 2005; Staal et al., 2003). In the modern Baltic
Sea a minimum temperature of 16°C is considered essential to initiate cHABs during summer,
when other crucial factors like low DIN/DIP ratio, calm winds and high irradiance occur
simultaneously (Kanoshina et al., 2003; Kononen, 1992; Kononen et al., 1996; Paerl, 2008;
Wasmund, 1997).
It should also be noted, however, that the homogeneous appearance of the sediments
and the much reduced TOC content (Fig. 3c) reveals that the water column was generally well
mixed and oxygenated during the Ancylus Lake and Yoldia Sea phases, resulting in a higher
degradation of organic matter (including HGs) in settling particles and surface sediments. To
compensate for this effect all HG concentrations were normalized to TOC content (Fig. 3a).
However, it is known that oxic conditions in the sediment result in a decreased preservation of

biomarkers relative to TOC (see Sinninghe Damsté et al., 2002). This may also explain in part the lower HG abundance in the Ancylus Lake and Yoldia Sea than in the brackish phase. However, it is noteworthy that no substantial change in the concentration of HGs is observed during the brackish phase when bottom water conditions changed from oxic to anoxic (Fig. 3). This suggest that the normalization to TOC content is an effective way to compensate for changing redox conditions of bottom and pore waters. The effect of oxic degradation is probably also not responsible for substantial changes in the distribution of the HGs since they are structurally similar and all contain a relatively labile glycosidic bond, so there is no reason to assume that one HG will degrade faster than another.

**Conclusions**

The distribution of the six analyzed $C_6$ HGs in the Baltic sediments from the brackish phases were closely related to those of cultivated heterocystous cyanobacteria of the family *Nostocaceae*. The record also shows that the HGs distribution has remained stable since the Baltic has turned into a brackish semi-enclosed basin ~7200 cal. yrs BP. During the freshwater phase of the Baltic (i.e. the Ancylus Lake phase) and an earlier brackish period (the Yoldia Sea phase) the distribution of the HGs was quite distinct but varied much more than in the subsequent brackish phase. This suggests that the cyanobacterial community adjusted to the different environmental conditions in the basin over this transition. We found that the abundance of HGs dropped substantially down-core, possibly either due to a decrease of the cHABs or during oxic degradation during deposition, resulting in partial destruction of the HGs.

In conclusion, it is likely that both salinity and temperature have influenced the abundance and composition of the heterocystous cyanobacterial community of the Baltic since the last deglaciation. The effects of salinity on the synthesis and distribution of HGs would need to be investigated in controlled conditions to be confirmed, as it has been partially done already in the case of temperature. Further studies are also needed to extend the range of heterocystous cyanobacteria species in culture that have been investigated for their HGs content.

**Acknowledgements**

We thank the captain and the crew of the R/V *"Prof. Albrecht Penck"* (cruise July 2009), and of the R/V *"Poseidon"* (cruise June 2012), for their support and assistance in the sampling.

We thank Mati Kahru for providing FCA data and three anonymous referees and Prof. Dan
Conley for helpful suggestions on an earlier draft of this paper. This project was funded by a
grant to JSSD from the Darwin Center for Biogeosciences (project nr. 3012). The work was
further supported by funding from the Netherlands Earth System Science Center (NESSC)
through a gravitation grant (NWO 024.002.001) from the Dutch Ministry for Education,
Culture and Science to JSSD.

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

**Table 1.** Distribution of the six targeted HGs in sediments from this study and from cultures of selected heterocystous cyanobacteria. Key: (++) Dominant (>25%); (+) Minor presence (5-25%; (tr.) Traces (<5%); (–) Not detected or not reported. Underlined strains were isolated from the Baltic Sea.

| Baltic Sediment | | $C_{26}$ diol | $C_{26}$ keto-ol | $C_{28}$ diol | $C_{28}$ keto-ol | $C_{28}$ triol | $C_{28}$ keto-diol |
|---|---|---|---|---|---|---|---|
| MoWP | | ++ | + | tr. | tr. | + | + |
| Pre-MoWP brackish | | ++ | +/tr. | + | tr. | +/tr. | tr. |
| Ancylus Lake-II | | ++ | +/tr | ++/+ | tr/- | tr/- | - |
| Ancylus Lake-I | | ++ | ++/+ | ++/- | ++/tr | +/tr | +/tr |
| Yoldia Sea | | ++ | + | ++/+ | +/tr | +/tr | tr |
| **Nostocaceae cultures** | **Strain ID** | | | | | | |
| *Nodularia* sp. [a] | CCY 9414 & 9416 | ++ | + | - | - | - | - |
| *Nodularia* sp. [b] | BY1 | ++ | + | - | - | - | - |
| *Nodularia* sp. [b] | F81 | ++ | + | - | - | - | - |
| *Nodularia* sp. [b] | AV1 | ++ | + | - | - | - | - |
| *Nodularia* sp. [b] | HEM | ++ | + | - | - | - | - |
| *Nodularia chucula* [a] | CCY 0103 | ++ | + | - | - | - | - |
| *Aphanizomenon* sp. [a] | CCY 0368 | ++ | + | + | - | - | - |
| *Aphanizomenon* sp. [a] | CCY 9905 | ++ | + | + | + | tr. | tr. |
| *Aphanizomenon* sp. [b] | TR183 | ++ | + | tr. | - | + | tr. |
| *A. aphanizomenoides* [c] | UAM 523 | + | - | ++ | + | tr | - |
| *A. gracile* [c] | UAM 521 | ++ | ++ | tr. | - | tr. | - |
| *A. ovalisporum* [c,f] | UAM 290 | ++ | tr. | tr. | tr. | - | - |
| *Anabaena* sp. [a] | CCY 0017, 9910, | ++ | + | + | + | - | - |
| *Anabaena* sp. [a] | CCY 9402 | - | - | ++ | + | - | - |
| *Anabaena* sp. [a] | CCY 9613 | + | + | - | - | - | - |
| *Anabaena* sp. [a] | CCY 9614, 9922 | ++ | + | - | - | - | - |
| *Anabaena* sp. [b] | 315 | ++ | ++ | tr. | tr. | tr. | - |
| *Anabaena* sp. [b] | BIR53 | ++ | ++ | tr. | tr. | tr. | - |
| *Anabaena* sp. [b] | BIR169 | ++ | + | tr. | tr. | ++ | + |
| *Anabaena cylindrica* [a] | CCY 9921 | ++ | + | - | - | - | - |
| *Anabaenopsis* sp. [a] | CCY 0520 | ++ | + | + | - | - | - |
| *Nostoc* sp. [a] | CCY 0012, 9926 | ++ | + | - | - | - | - |
| *Nostoc* sp. [c] | MA 4 | ++ | ++ | tr. | - | tr. | - |
| *Cylindrospermopsis raciborskii* [c,f] | UAM 520 | ++ | tr. | + | tr. | + | - |
| *Cyanospira rippkae* [e] | ATCC 43194 | - | - | ++ | + | - | - |
| **Rivulariaceae cultures** | | | | | | | |
| *Calothrix desertica* [d] | PCC 7102 | - | - | - | - | ++ | ++ |
| *Calothrix* sp. [c] | MU 27 | - | - | tr. | - | ++ | ++ |
| *Calothrix* sp. [a] | CCY 0018 | - | - | - | - | ++ | + |
| *Calothrix* sp. [a] | CCY 0202 | tr. | tr. | - | - | ++ | + |
| *Calothrix* sp. [a] | CCY 0327 | - | - | - | - | ++ | + |
| *Calothrix* sp. [a] | CCY 9923 | - | - | + | + | ++ | + |
| **Microchaetaceae cultures** | | | | | | | |
| *Microchaete* sp. [d,f] | PCC 7126 | - | - | + | ++ | - | - |
| **Tolypothrichaceae cultures** | | | | | | | |
| *Tolypothrix tenuis* [d,f] | PCC 7101 | - | - | ++ | + | - | - |

a = Bauersachs et al. (2009a), b = Bauersachs et al. (2017). c = Wörmer et al. (2012), d = Gambacorta et al. (1998), e = Soriente et al. (1993). f = these species also contain HGs other than the six HGs targeted in this study

28 **Table 2.** Analysis of HGs in Baltic sediments by Orbitrap MS. Key: (++) Dominant (>25%); (+) Minor presence (5-25%; (tr.) Traces (<5%); (−)
29 Not detected. Relative abundances are based on peak areas.

| Sample | C5 26 diol | Deoxy C6 26 diol | C6 26 diol[a] | C6 26 keto-ol | C6 28 diol | C6 28 keto-ol | C6 28 triol | C6 28 keto-diol | C6 30 triol | C6 30 keto-diol | C6 32 triol | C6 32 keto-diol |
|---|---|---|---|---|---|---|---|---|---|---|---|---|
| P435-1-4 MUC 4 | - | - | ++ | + | tr. | tr. | ++ | + | - | - | - | - |
| P435-1-4 MUC 35 | - | - | ++ | + | tr. | - | + | + | - | - | - | - |
| P435-1-4 MUC 62 | - | - | ++ | + | tr. | - | ++ | + | - | - | - | - |
| P435-1-4 MUC 99 | - | - | ++ | tr. | tr. | - | ++ | tr | - | - | - | - |
| GC 1-2 cm | - | - | ++ | tr. | tr. | - | + | - | - | - | - | - |
| GC 5-6 cm | - | - | ++ | tr. | tr. | - | + | - | - | - | - | - |
| GC 17-18 cm | - | - | ++ | tr. | tr. | - | + | - | - | - | - | - |

[a] sum of two isomers

**Figure legends**

**Figure 1.** Structures of the $C_6$ heterocyst glycolipids (HG) targeted by the study. $C_{26}$ diol HG (1-(O-hexose)-3,25-hexacosanediol); $C_{26}$ keto-ol HG (1-(O-hexose)-3-keto-25-hexacosanol); $C_{28}$ diol HG (1-(O-hexose)-3,27-octacosanediol); $C_{28}$ keto-ol HG (1-(O-hexose)-3-keto-27-octacosanol); $C_{28}$ triol HG (1-(O-hexose)-3,25,27-octacosanetriol); $C_{28}$ keto-diol HG (1-(O-hexose)-27-keto-3,25-octacosanediol).

**Figure 2.** Map of the Baltic Sea. The location of multicore (MUC) P435-1-4 and gravity core (GC) 303600 in the eastern Gotland Basin is indicated with a black star (modified from Warden et al., 2017).

**Figure 3.** Proxy records of the Baltic Sea cores on a composite depth scale aligned with core photos showing the lamination of the sediments of the post-Ancylus Lake stage. (**a**) The abundance of the HGs (r.u. gTOC$^{-1}$) on a log scale, (**b**) $\delta^{15}N$ (‰), (**c**) TOC content (%) partly derived from Warden et al. (2017), and (**d**) TEX$_{86}$-derived summer sea surface temperatures (SSTs) from Kabel et al. (2012) and Warden et al. (2017). Data points derived from the MUC P435-1-4 core are in grey and those from the GC 303600 core are in black. The stratigraphy is based on age models published elsewhere (Kabel et al., 2012; Warden et al., 2017) and for the deeper part of the GC 303600 core on unpublished data on diatom assemblages. The TEX$_{86}$ data of Kabel et al. (2012) were measured on a different core (MUC 303600) obtained from the same site, which was correlated to the MUC P435-1-4 core based on the TOC profiles (Fig. S1). Note that phases characterized by deposition of laminated sediments are the periods during the Holocene when the bottom waters of the Baltic Sea were anoxic.

**Figure 4.** Distribution of HGs, displayed as fractional abundance (%), versus depth (in cm) for (**a**) the MUC P435-1-4 core, and (**b**) the GC 303600 core. Colour key: light green: $C_{28}$ triol HG; blue: $C_{28}$ keto-diol HG; yellow: $C_{28}$ diol HG; orange: $C_{28}$ keto-ol HG; purple: $C_{26}$ diol HG; red: $C_{26}$ keto-ol HG. Each sample represents a sediment slice of 0.5 cm in the case of the MUC and of 1 or 2 cm in the case of the GC. The stratigraphy of the cores (see Fig. 3) is indicated. The scores on PC1 and PC2 derived from the principal component analysis of the HG distribution are plotted along the fractional abundance plots using the same scale for both cores.

**Figure 5.** Principal component analysis of the heterocyst glycolipids (HGs) distribution in the
sediments recovered by the MUC P435-1-4 and the GC 303600 cores from the Gotland Basin,
Baltic Sea. (**a**) The loadings of the six HGs on the first two PCs, with PC1 accounting for the
47% and PC2 for 29% of the variance. (**b**) Scores of the sediments from various stages on
PC1 and PC2.
**Figure 6**. Abundance of heterocyst glycolipids (HG) in the Baltic Sea over the period 1977–
2012 (from MUC) compared with the fractional cyanobacteria accumulation (FCA, %) from
the time period 1979–2012, as reported by Kahru et al. (2014).

Figure 1

C$_{26}$ diol

C$_{26}$ keto-ol

C$_{28}$ diol

C$_{28}$ keto-ol

C$_{28}$ triol

C$_{28}$ keto-diol

# Figure 2

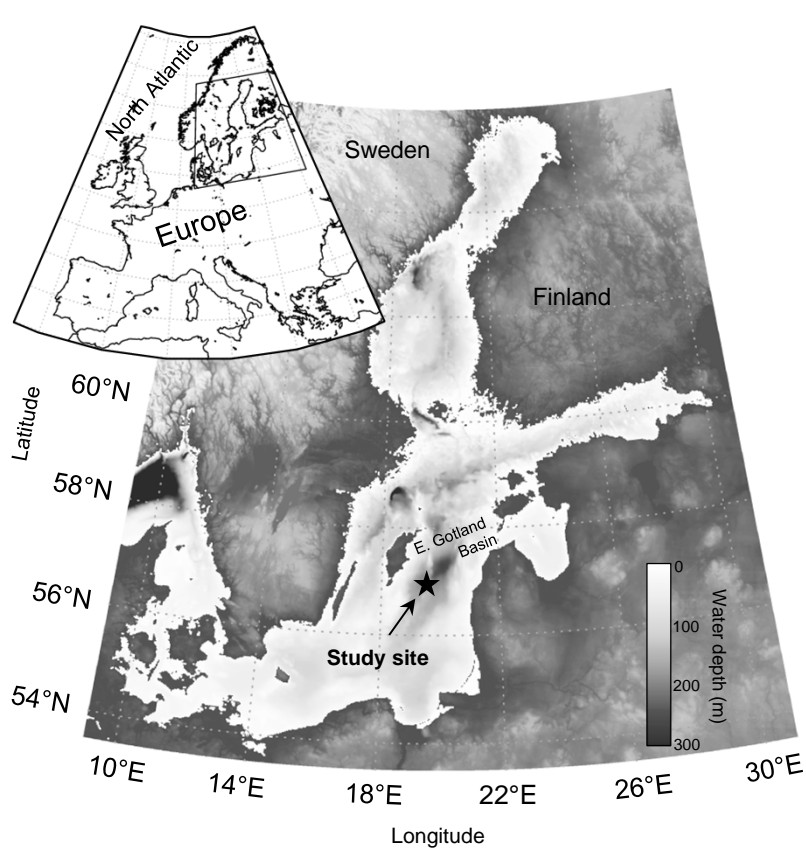

Figure 3

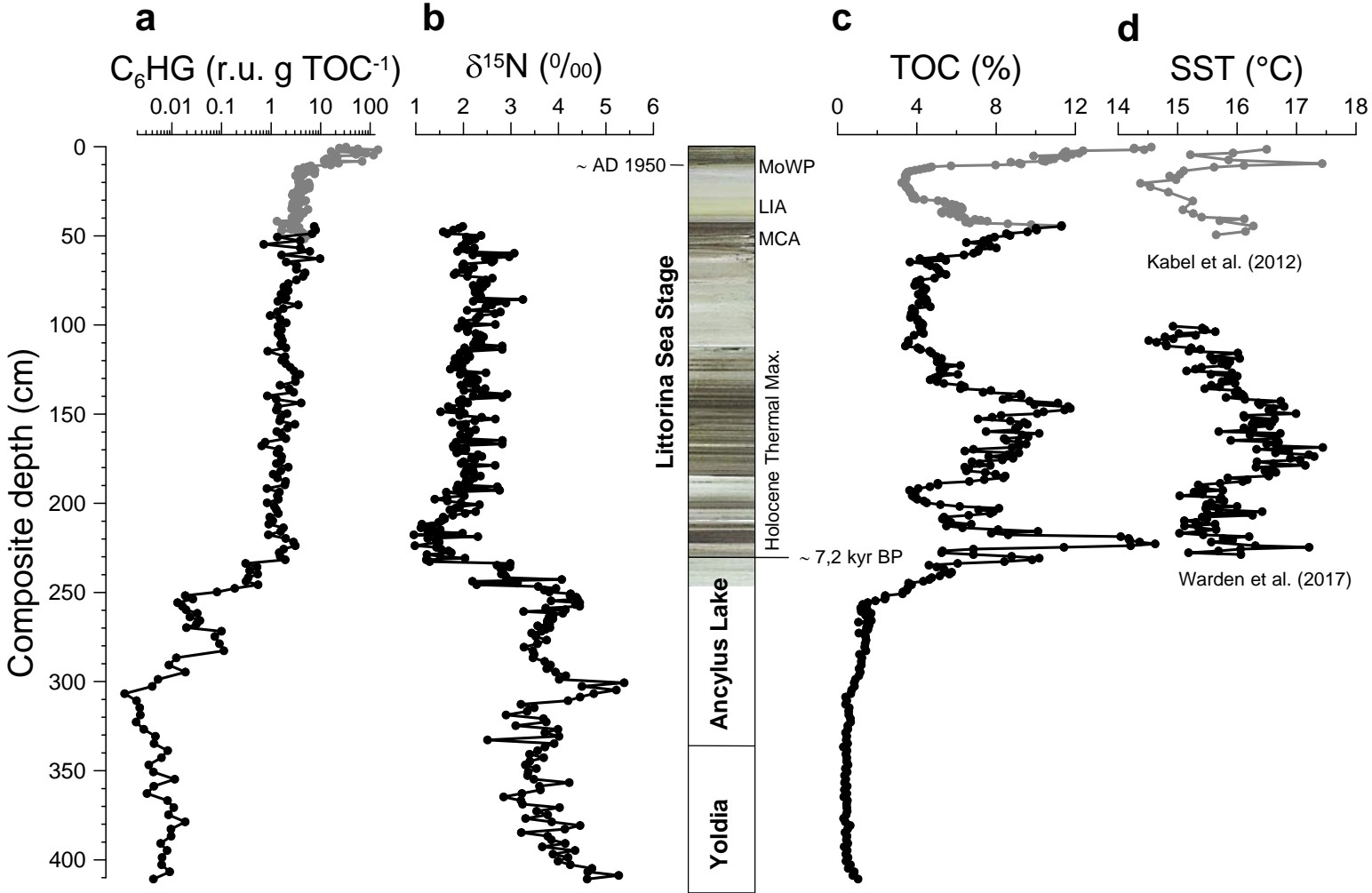

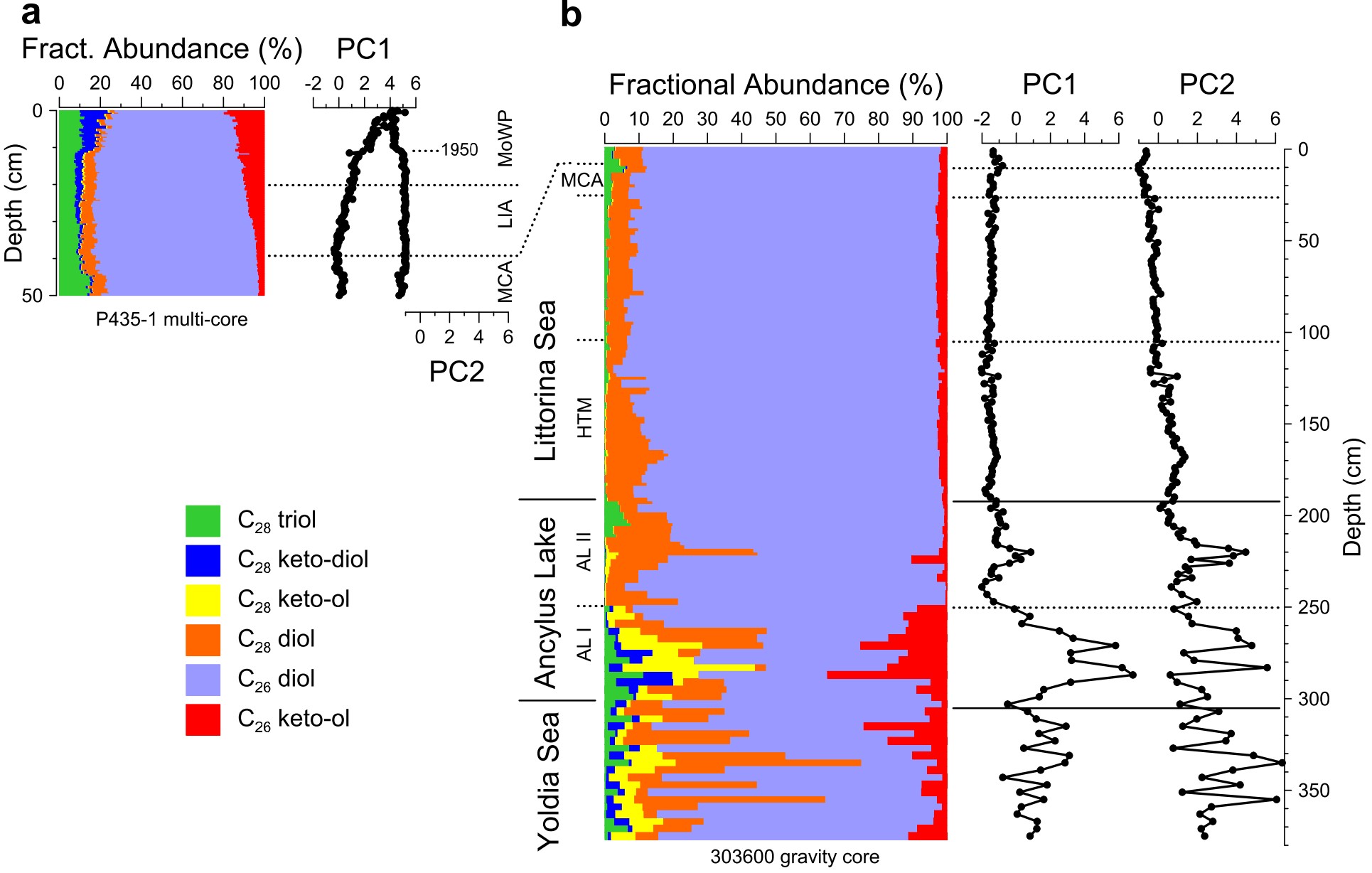

Figure 4

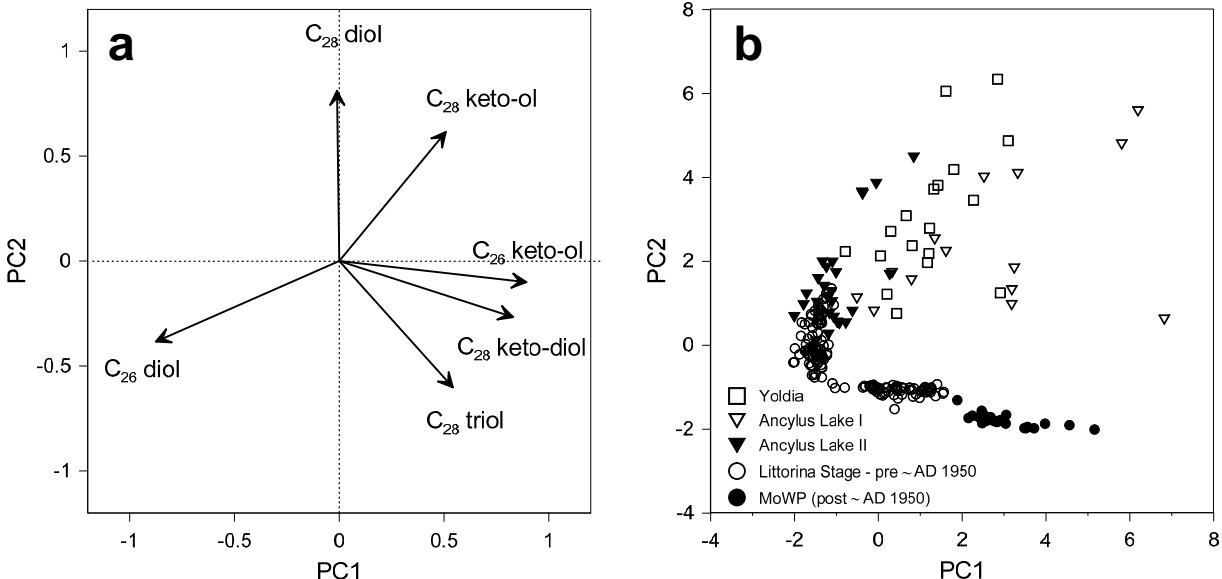

Figure 5

# Figure 6

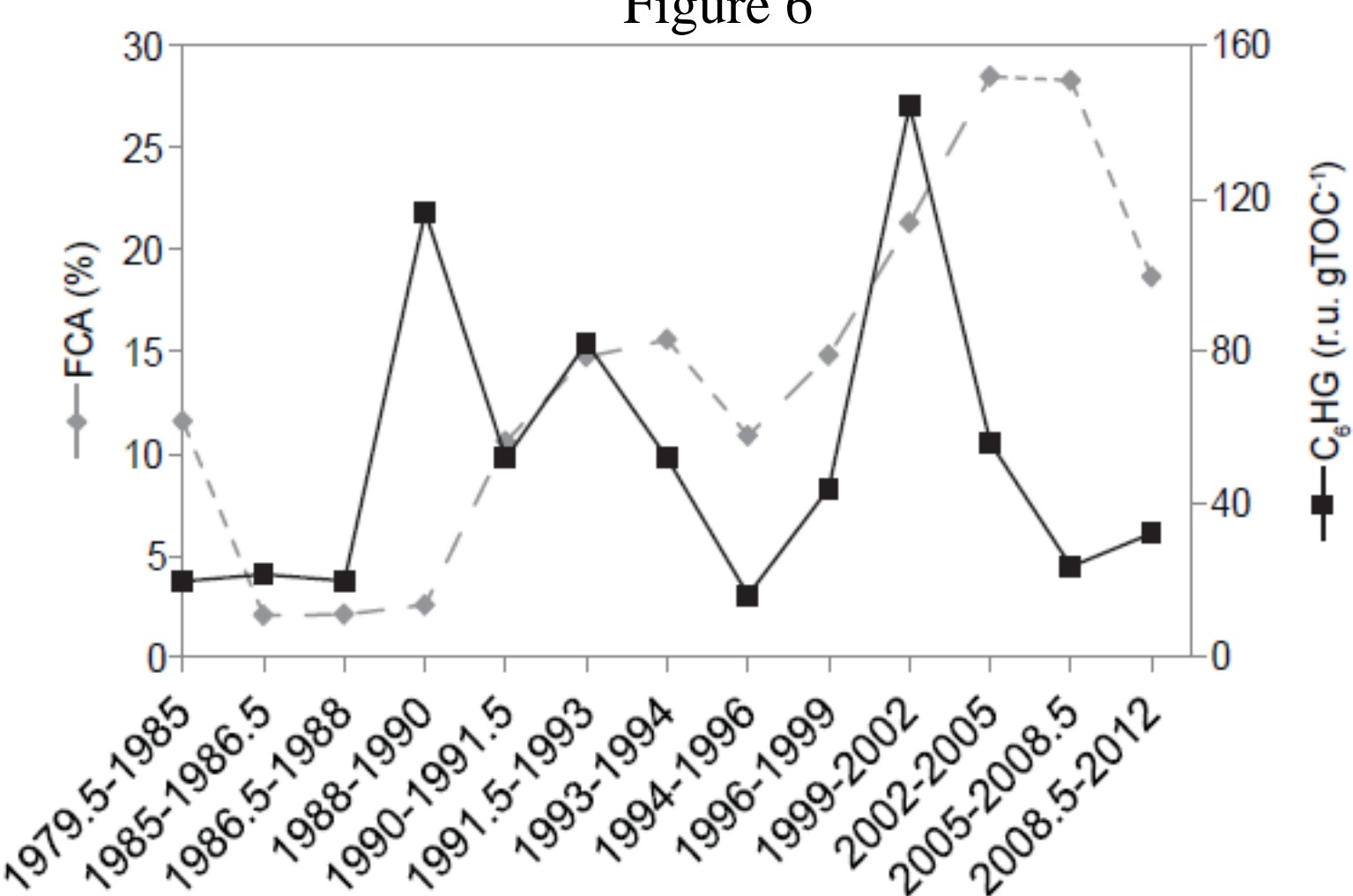

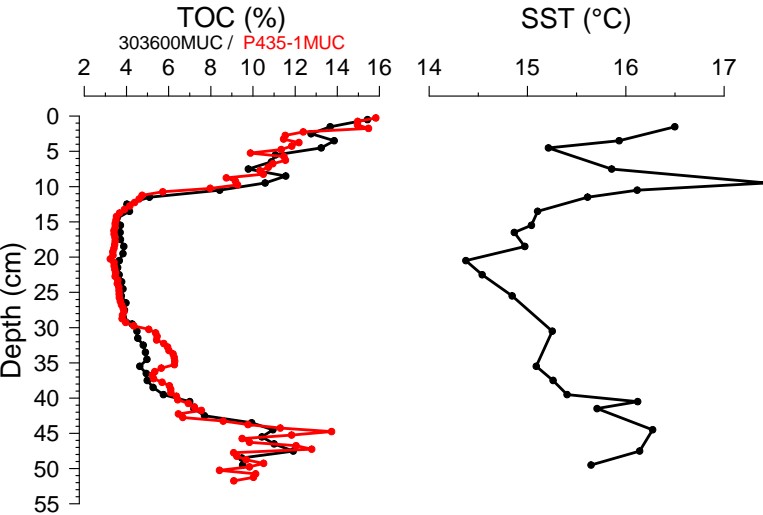

Figure S1: Correlation of the MUC cores 303600 (Kabel et al., 2012) and P435-1 (this work) on the basis of the TOC content. SST data is from MUC 303600 (Kabel et al., 2012).