# Peer review of "The Holocene sedimentary record of cyanobacterial glycolipids in the Baltic Sea: Evaluation of their application as tracers of past nitrogen fixation"

_Biogeosciences, 2017_

## Referee Comment (RC1) · Anonymous Referee #1 · 30 Aug 2017

This paper reports on the downcore distribution of heterocyst glycolipids (HGs) in the Baltic Sea in an attempt to evaluate the utility of HGs as tracers for past nitrogen fixation by cyanobacteria. The rationale is very well formulated and the data are unique and precious given the high-resolution sedimentary record and the limited information on these biomarkers in paleo-environment studies. I am very impressed with the depth of analytical analysis involved and the motivation of research. However, I must admit that I am not convinced that the data deliver the conclusion described in the abstract. I have two major concerns.

[Figure]

My first concern relates to the preservation of HGs in sediments (as is briefly discussed by the authors in the text as well). How does HG decomposition vary in freshwater versus brackish water systems? In modern freshwater and brackish water systems, does HG composition show the same pattern as observed in the sediment core? Is it possible that HGs are better preserved in brackish waters, leading to their higher abundance as well as stability compared to in freshwater systems? If so, HGs in sediments are not only related to their inputs but also to their decay. As both processes are influenced by temperature, the presence of O2 and possibly salinity, it is very difficult to conclude on "the potential of HGs as specific biomarker of heterocystous cyanobacteria in paleo-environmental studies". Instead, I would suggest considering whether there is a proxy or indicator that may be used to (even roughly) assess the preservation or degradation stage of HGs in sediments? In lines 27-35 (pg 9), it is mentioned that sea surface temperatures reconstructed using HGs were too high to be realistic and the causes were not clarified. To me, this seems like a hint that HG signatures in the sediments may be subject to diagenesis-related alterations and that different molecules have be influenced differentially. I think the authors need to clarify this possibility before making conclusions and in the abstract as well.

My second concern relates to the influence of multiple environmental variables on HG composition and distributions. As the authors mentioned (several times) in the text, HG variations may be related to temperature variations as well as salinity changes. I think that control experiments are needed to prove that HG shifts are related to cyanobacteria community changes only instead of being affected by physiochemical processes also.

A minor point: I am not sure if Figure 6 provides any new information—this is just another form of Figure 4. The authors may consider removing this figure.

---

## Author Comment (AC1) · 14 Sep 2017

We thank referee #1 for taking the time to read and comment on our manuscript. We do appreciate the positive assessment of our work. Here we respond to the various issues raised in the two concerns of the referee. - Concern #1: "My first concern relates to the preservation of HGs in sediments (as is briefly discussed by the authors in the text as well). How does HG decomposition vary in freshwater versus brackish water systems? In modern freshwater and brackish water systems, does HG composition show the same pattern as observed in the sediment core? Is it possible that HGs are

better preserved in brackish waters, leading to their higher abundance as well as stability compared to in freshwater systems? If so, HGs in sediments are not only related to their inputs but also to their decay. As both processes are influenced by temperature, the presence of O2 and possibly salinity, it is very difficult to conclude on "the potential of HGs as specific biomarker of heterocystous cyanobacteria in paleoenvironmental studies". Instead, I would suggest considering whether there is a proxy or indicator that may be used to (even roughly) assess the preservation or degradation stage of HGs in sediments? In lines 27-35 (pg 9), it is mentioned that sea surface temperatures reconstructed using HGs were too high to be realistic and the causes were not clarified. To me, this seems like a hint that HG signatures in the sediments may be subject to diagenesis-related alterations and that different molecules have be influenced differentially. I think the authors need to clarify this possibility before making conclusions and in the abstract as well."

This concern of the referee relates to two fundamental questions: what is the effect of changes in preservation conditions on 1) the concentration and 2) the distribution of HGs in sediments. Indeed the ability of HGs to be preserved in sediments represents a key premise to our work (as it is for every biomarker). We discuss potential break-down of HGs in sediments quite extensively (pages 10–11, lines 30–40 and 1–22) but this is mainly related to the breakdown in the anoxic sediments (i.e. the marked decline in the concentration of HGs in uppermost sediments). We have interpreted their decline in the sediments deposited during the freshwater phase as a much lower abundance of nitrogen-fixing cyanobacteria (which is supported by the increased $\delta 15N$ values) but it is true that we should have discussed more extensively the fact that the changing redox conditions in the surface sediments (i.e. from anoxic to oxic) will probably have affected the conditions for preservation of HGs. To compensate for this we normalized the HG concentrations on TOC but it is known that biomarkers are more readily degraded than TOC under oxic conditions. In the revised version of our manuscript we will discuss this topic more in depth. On the matter of how the HG decomposition varies in modern freshwater versus brackish water systems (i.e. effect of salinity), to the best

of our knowledge, no studies are available. However, we think that other environmental parameters (e.g. oxygen exposure) are far more important. The second issue that was raised concerns the question whether perhaps partial degradation of HGs results in significant changes in the distribution of the HGs in such a way that differences in HG distribution, interpreted as arising from a difference in the composition of heterocystous cyanobacteria (e.g. as done in our study for the differences observed between the Ancylus Lake and brackish Baltic Sea), is in fact caused by differences in degradation of individual HGs. We don't feel that this is likely. First of all, all HGs are chemically quite similar and we don't expect large differences in oxic degradation rates. Secondly, in a number of systems a good match in the distribution of the HG of suspended water column and surface and (in some cases) deeper sediments is observed (i.e. Lake Challa, Bauersachs et al 2010; Lake Schreventeich, Bauersachs et al., 2015; equatorial Atlantic, Bale et al., 2017). Upon sedimentation, a substantial fraction of the HGs will be degraded, so these studies indicate no preferential degradation of specific HGs. Lastly, the HG distribution that we find in the surface sediments of the Baltic Sea are fully in line with the HG composition of the most important heterocystous cyanobacteria (see text of the manuscript). Consequently, we don't see this as a problem and have full confidence in using the HG distribution to infer potential sources but we will elaborate on this a little bit more in the revised version of our manuscript.

- Concern #2: "My second concern relates to the influence of multiple environmental variables on HG composition and distributions. As the authors mentioned (several times) in the text, HG variations may be related to temperature variations as well as salinity changes. I think that control experiments are needed to prove that HG shifts are related to cyanobacteria community changes only instead of being affected by physiochemical processes also."

We do agree with the suggestion that controlled experiments would help to further elucidate the influence of environmental factors such as temperature and/or salinity on the HG composition. As indicated by the referee, we already quote quite a number of

studies that have examined the influence of temperature and these studies have even resulted in the potential application of HGs to reconstruct temperature. Studies on the effect of salinity on the HG composition of different heterocystous cyanobacteria have not been performed but are more difficult because of the restricted salinity range of heterocystous cyanobacteria. However, these kind of studies fall outside the scope of the present work that describes the HG composition of the Holocene sedimentary record of the Baltic Sea.

References

Bauersachs, T., Speelman, E. N., Hopmans, E. C., Reichart, G. J., Schouten, S., & Sinninghe Damsté, J. S. (2010). Fossilized glycolipids reveal past oceanic N2 fixation by heterocystous cyanobacteria. Proceedings of the National Academy of Sciences, 107, 19190-19194.

Bauersachs, T., Rochelmeier, J., & Schwark, L. (2015). Seasonal lake surface water temperature trends reflected by heterocyst glycolipid-based molecular thermometers. Biogeosciences, 12, 3741-3751

Bale N.J., Villareal T.A., Hopmans E.C., Brussaard C.P.D., Besseling M., Dorhout D., Sinninghe Damsté J.S. and Schouten S. (2017). C5 glycolipids of heterocystous cyanobacteria track symbiont abundance in the diatom Hemiaulus hauckii across the tropical north Atlantic. Biogeosciences Discussions, https://doi.org/10.5194/bg-2017-300

---

## Referee Comment (RC2) · Anonymous Referee #2 · 15 Sep 2017

This manuscript is a substantial contribution to developing a molecular proxy for N2-fixation in the geological record, namely the diagnostic glycolipids indicative of hete-rocyst envelopes. The manuscript is very well written, structured, and points are well argued. The amount of analyses is staggering and I definitely support publication of the paper. An impressive set of analyses of these heterocyst glycolipids (HGs) in dated sediment cores from the Baltic Sea are the basis on which the authors explore some very interesting ideas: The modern Baltic Sea is known for massive blooms of two species of cyanobacteria and there is evidence from molecular and isotope data that

they occurred during much of the Littorina Sea (LS) Stage and subsequent brackish phases. Whether N-fixation was a feature in the pre-Littorina lacustrine stages was unknown (at least to me). A first objective was to test if the HG patterns give evidence for alternating communities mirrored in HG distribution patterns. That appears indeed to be the case. HG patterns have been remarkably stable over the last 7000 years or so, although HG abundances in the short core show a typical decrease in their contribution to total organic carbon (TOC) that suggests that they are more rapidly degraded than bulk TOC (Fig. 6 upper left). Below 200 cm in the longer core, both abundances relative to TOC and HG composition are highly variable. These variations characterize the lacustrine Ancylus Lake (AL) Stage, that must have received input of organic matter containing HGs. Here are my first questions: What are the HG patterns of soil cyanobacteria, and is the input of soil-derived TOC a possible source and also possibly a reason for differences in AL and LS sediments? I seem to remember that lignin biomarker abundance increased at the AL/LS transition. What are the levels of r.u. compared to other depositional settings? Is the Baltic Sea particularly rich in HGs? A second (and interesting) objective was to investigate a fundamental biogeochemical feedback: Because the brackish Baltic Sea (LS and younger stages) experienced several alternations between oxic and anoxic conditions, it is a well-chosen environment to investigate whether or not development of anoxia and the Redfield homeostat (nitrogen fixation balancing a surplus of P originating from sediment or from denitrification) are linked, and if cyanobacterial biomass has an influence on the development of anoxia (or if anoxia had an influence of HG production). This is a difficult question and I wonder if it can be answered at all if you normalize your r.u. to %TOC. Are the unnormalised r.u. linearly correlated with %TOC? Figure 3a in comparison to 3 e suggests this. That would mean that TOC preserved is the overriding control on HG abundances (but not composition) – by normalizing to TOC, any variation in HG abundance will then be masked. If TOC is high in anoxic and low in oxic phases, the effects of production and preservation can in my opinion not be segregated. Does the downcore decrease in PC1 in the MUC mean that the HG are more labile than bulk TOC? In particular,

the relative abundances in Figure 4 suggest to me that C28 keto-diol and C26 keto-ol must be more rapidly degraded that the other moieties. Have you analysed the principal components for the MUC and GC separately, and are the score patterns similar to those for the entire sample pool?

The following are some queries, remarks and details Why do some labs continue to use acidified samples for d15N analyses in the face of ample evidence that this affects the values? But that is not crucial to this paper. Page 10 L14-29 and Figure 7: What is the correlation coefficient of the two data series? Is there an estimate of how much of the present-day cyanobacterial detritus reaches the sea floor in comparison to biomass produced? P3 L2: what is "fully brackish"? Figure 4: Labels are too small; Y-axis unit left graph must be "cm" Page 5 L16: delete ")" P9 L7: delete "permil" after salinity value

---

## Short Comment (SC1) · 25 Sep 2017

I just have two brief comments regarding this manuscript.

1) It appears that the 15N samples were acidified before measurement, which has been shown in the literature to result in anomalous values. If you compare their 15N data with other data from the Baltic Sea from a variety of groups their data show very little variation through time especially during hypoxic periods.

2) The HG data also show a different picture than what has been observed with pigment

biomarkers for cyanobacteria in the Baltic Sea. Funkey et al. (2014) – is referenced, but not discussed - showed increased cyanobacteria abundance during period of hypoxia likely due to changes in the biogeochemistry of P during low oxygen periods.

I think more needs to be done to assure the validity of the 15N measurements and other proxies should be measured and compared to validate the HG data.

Daniel Conley, Department of Geology, Lund University

---

## Short Comment (SC2) · 29 Sep 2017

1) Point taken. However, the pattern is largely obscured in the narrow plot. If you increase the width of the plot you can see variations in the 15N data. Lower values of 15N can be indicative of nitrogen fixation and are considerably lower during the transition from the Ancylus Lake to establishment of the Littorina Sea, although that is not reflected in the HG plot. In addition, during periods of laminated sediments the 15N is ca 1 per mil lower consistent with Funkey et al. Perhaps replotting the data would be an option?

2) I agree that all biomarkers are not created equal, but I also know that it is more powerful to use a suite of biomarkers to identify changes in biogeochemistry. Throughout most of the Funkey et al. core from the Baltic Sea pheophytin-a/Chl-a was relatively constant (except in the upper core) suggesting no large changes in diagenesis. And yes, zeaxanthin occurs in many organisms, but it is widely used as one of the biomarkers for cyanobacteria in aquatic ecosystems. For example, Bianchi et al. 2000 showed changes in zeaxanthin in laminated sequences in the Baltic in addition to changes in TC (which your paper also shows) and changes in the N:P ratio. Bianchi et al. also show a large increase in zeaxanthin/B,B-carotene ratio at the Ancylus/LS transition, which is seen in your HC data. Echinenone is highly diagnostic appearing in only cyanobacteria and sea urchins and confirms the zeaxanthin concentrations.

I am still surprised that the HG data does not show variations in cyanobacteria abundance. I saw a talk at Goldschimdt 2017 on the Baltic Sea, probably unpublished data, that confirms the Funkey et al. hypothesis. That Sollai et al. did not observe changes in HC is interesting and that it is different from Funkey et al. and Bianchi et al. is also interesting – that's science. But I think it warrants addressing in the Sollai et al. paper.

---

## Short Comment (SC3) · 29 Sep 2017

Okay, we will re-plot del15N data and discuss the discrepancy with the, in my view, less specific carotenoids data published earlier.

---

## Referee Comment (RC3) · Anonymous Referee #3 · 17 Oct 2017

The manuscript reports on the use of lipids specific to heterocystous cyanobacteria (heterocyst glycolipids) as (i) tracers for investigating past changes in the community of cyanobacterial blooms and (ii) paleo-proxy to trace back anoxic events in the Holocene Baltic Sea. Sediments sampled from a multicore and a gravity core collected in the Gotland Basin have been investigated for bulk geochemistry, nitrogen isotopes and the distribution and abundance of heterocyst glycolipids. While the use of heterocyst gly-colipids as biomarkers to trace for cyanobacterial blooms in the Baltic Sea is principally interesting, I have some major concerns regarding the experimental setup, study de-

sign as well as data acquisition and interpretation that have to be addressed before I can recommend publication of the manuscript.

General comments

My first and most pressing concern is related to the reconstruction of the past Baltic Sea cyanobacterial community, which seems to be the major aims of the study. From reading the manuscript, I got the impression that only six C6 HGs were present in the Baltic Sea sediments and that those are indicative mainly for heterocystous cyanobacteria of the family Nostocaceae; in agreement with the major bloom-forming Baltic Sea cyanobacteria. However, while having a closer look at the method used for the detection of HGs (apparently the same method described by Bale et al. (2015; OG)), I could not fail to notice that the method specifically targets only these six C6 HG but it is neither able to detect HGs of longer chain length (e.g. C30 to C32 keto-ol, keto-diol, diol and triol HGs), which have been described from numerous heterocystous cyanobacteria previously (Gambacorta et al. 1998; Phytochemistry) nor HGs with deoxyhexose or pentose headgroup (attached to a C26 alkyl chain) as described by Wörmer et al. (2012; L&O). This essentially means that the authors limit themselves to a very narrow window of HGs and consequently members of the cyanobacterial community that can be detected with their method. Moreover, they limit themselves largely to the detection of Nostocaceae. So, my major concern is: do the presented HG profiles really reflect the complete cyanobacterial community or in fact only a small fraction of the community and is it then possible to draw any conclusion on the cyanobacterial community at all? It is very much likely that HGs with higher carbon chain length or other sugar head groups are also abundant and perhaps also more dominant than the six C6 HGs that were detected in the Baltic Sea sediments but we would never know because they are not included in the detection method. This might be in particular the case for the freshwater interval, for which major changes in the cyanobacterial community would be expected.

Moreover, it makes of course sense that the presented HG profiles agree with the major

bloom-forming genera if only those HGs are included in the detection method that are specific for cyanobacteria of the family Nostocaceae. In my opinion, the authors may have missed major changes in the cyanobacterial community due to the limited number of HGs that have been investigated. To obtain robust results and hence make reliable interpretations of cyanobacterial community changes over time, most if not all of the samples would need to be re-measured using a method that includes the full spectrum of HGs currently known from heterocystous cyanobacteria.

This identification of potential biological sources of HGs in the Baltic Sea sediments is similarly problematic. I got again the impression that the six C6 HGs shown in Table 2 and discussed in the text cover the full spectrum of HGs that are present in the listed heterocystous cyanobacteria. From reading the original literature, however, it seems that many of these species do not only contain the six C6 HGs but also other HGs in substantial abundances, in particular when they do not belong to the Nostocaceae. For example, according to the authors, Tolypothrix contains only C28 diol and keto-ol HGs but in fact it also contains significant quantities of C30 triol and keto-diol HGs that surprisingly have not been included in the table and again this component will be missed in the Baltic Sea sediments as it is not included in the detection method. Likewise, according to the authors the C28 triol HG should be the only HG present in Scytonema hofmanni. This by no means is the case if the original literature is consulted (Gambacorta et al. (1998; Phytochemistry)). In fact, this HG is not present in S. hofmanni at all. Instead, it only contains C30 triol and keto-diol HGs, both of which cannot be detected using the analytical protocol described in the present study. There are other examples, such as Aphanizomenon aphanizomenoides or Aphanizomenon ovalisporum, from which only incomplete HG profiles are described lacking e.g. C30 diol, triol, keto-ol and keto-diol HGs as well as HGs with deoxyhexose or pentose head group. I am wondering why only a selection of HGs is shown in the table and why this is biased towards Nostocaceae? In any case, the question remains: Can the cyanobacterial community reconstructed based on these incomplete records? I think not. Therefore, I strongly encourage the authors to carefully check the table and where

necessary to complete the full range of HGs. Otherwise, it is not possible to link HG profiles detected in the sediment record to the biological sources of HG and any attempt to reconstruct cyanobacterial community changes will be flawed.

Although I generally appreciate the authors' efforts to identify the sources of HGs in Baltic Sea sediments, I have my doubts that this is possible by comparing sedimentary HG profiles with HG distributions in cultured cyanobacteria. All cyanobacteria investigated for their HG content so far include either freshwater or marine representatives but brackish species (such as those from the Baltic Sea) have not been analyzed so far. Given that the environmental conditions in the Baltic Sea significantly differ from freshwater and marine environments, it is likely that Nodularia, Aphanizomenon or Anabaena species living in the brackish Baltic Sea will not necessarily show similar HG profiles as found in freshwater and marine cyanobacteria. Although it requires additional work, the authors may consider including modern Baltic Sea cyanobacteria in their study, so that HG distributions can be unequivocally linked to their biological sources and eventually be used to reconstruct changes in the cyanobacterial community in the Holocene Baltic Sea. This would significantly strengthen their conclusions.

While having a look at the HG structures, I am wondering if accelerated solvent extraction is the method of choice for extracting HGs from sediments? For my feeling, this particular extraction method is too harsh and may lead to the degradation of HGs. I assume that there is a reason why other studies dealing with HGs (such as Bale et al. 2015 (OG), 2016 (L&O); Schouten et al. 2013 (Phytochemistry); Bauersachs et al. 2009 (Phytochemistry); 2015 (Biogeosciences)) have used the more gentle Bligh and Dyer extraction method? Can the authors proof beyond doubt that the extraction method did not flaw the generated HG profiles and that these profiles are indeed representatives for the sedimentary signal? While reading some of the original literature, I noticed that a comparison between ASE and Bligh & Dyer extraction has been made previously (Bauersachs et al. 2010; PNAS). Yet, the comparison is only semi-quantitative and without quantification using a standard also less robust. From these

experiments it is also not clear whether ASE leads to selective degradation of keto-ol vs diol HGs or diol vs triol HGs. This is likely not an issue with Bligh and Dyer but with ASE it may indeed be problematic. This issue, however, it is not addressed in the manuscript. As it is now possible to quantify HGs (see Bale et al. 2016; OG), these experiments could easily be done and included in the manuscript.

The issue of how degradation may affect HGs is also only little addressed but essential to verify the robustness of these components as biomarkers for cyanobacterial HABs in the Baltic Sea and as paleo-proxies. As stressed by the other reviewers, the HG profile does not really match other profiles of cyanobacterial activity such as those based on cyanobacterial pigments reported by Bianchi et al. (2000; L&O) or Funkey et al. (2014; EST). Yet, the nitrogen isotope record for instance shows lowest values at the AL-LS transition and the lower part of the Littorina Sea phase, which may point to an increased loading of nitrogen derived from cyanobacterial N2 fixation. Therefore, it is surprising that this interval is not characterized by increased abundances of HGs. As indicated in the text, heterocystous cyanobacteria may not have formed blooms in the past Baltic Sea but this is in contrast to previous findings and certainly needs more attention in the manuscript. It could also very well be that HGs experienced some sort of degradation and are therefore not abundant in the lower part of the Littorina Sea phase anymore. Determining the degradation of HGs is certainly beyond the scope of the manuscript but it would be interesting to obtain additional proof for the presence/absence of cyanobacteria along the record. One such proxy is pigments but distributions of methyl branched alkanes (a well-established marker for cyanobacteria) are an alternative. If these independent proxies show similar profiles as the HGs, this would at least strengthen the authors' hypothesis that cyanobacterial HABs played only a minor role in past Baltic Sea.

The use of the temperature indices does not seem to add much to the manuscript and I am wondering if it is really needed. Temperatures reconstructed using the HDI26 or HDI28 are not described in the results section and only briefly touched on in the discussion section (p. 9, l. 20-35). I also find this discussion hard to follow. It is not really clear to me how they calculated the temperatures. It is indicated that proxy calibrations from cultures were used but the calibrations described in the 'materials and methods' section seem to be those established by Bauersachs et al. (2015; Biogeosciences) for a lake environment. This is confusing and it should be clarified which calibration has been applied before any robust discussion of temperature can actually be made. I also have my doubts that the culture or the lake calibrations are indeed applicable in the brackish Baltic Sea and that the 'somewhat unrealistic' and too high temperatures result from the lack of a calibration specifically established for the Baltic Sea. There are numerous other examples where specific calibrations have been generated for the Baltic Sea including e.g. the TEXL86.

The publication would greatly benefit from including a temperature record based on a well-established temperature proxy, such e.g. the TEXL86. As mentioned on several occasions in the manuscript, 16 °C seems to be a sort of threshold with temperatures >16 °C promoting bloom-formation. If established and plotted along with the HG data, it would allow identifying intervals during which past cyanobacterial HABS may have occurred in the Baltic Sea.

Specific Corrections

p. 1, l. 17-18. Please mention the different genera of bloom-forming heterocystous cyanobacteria.

p. 2, l. 7. Adams was certainly not the first one to describe the role of the heterocyst in the process of N2 fixation. A very nice overview on this topic is provided by Wolk (1982) and this author certainly deserves credit for his work. Please add the following reference to the manuscript:

Wolk, CP (1982). Heterocysts. In: Carr, N.G., Whitton, B.A. (Eds), The Biology of Cyanobacteria. Blackwell Scientific Publishers, Oxford, pp. 359-386.

p. 2, l. 7-9. The way the sentence is phrased, it seems that both the polysaccharide and the glycolipid layer are involved in regulating the diffusion of atmospheric gases to the heterocyst. Yet, the polysaccharide layer is considered to provide protection with regard to mechanical damages. Please rephrase to make clear that the glycolipid layer is the gas diffusion barrier.

p. 2, l. 15: I do not like the term 'free-living' too much. I think 'non-symbiotic' is more appropriate in this context.

p. 2, l. 31. A reference to studies addressing the nature of cyanobacterial HABs is missing here.

p. 3, l. 8. In addition, species of the genus Anabaena may also be important bloom-formers and they should be included here. They are mentioned as bloom-formers in the discussion section. So why not here as well?

p. 3, l. 33. The authors state to investigate 'past cyanobacterial communities' but in fact they limit themselves to a very narrow range of the cyanobacterial community as their method only allows the detection of six C6 heterocyst glycolipids. As expressed in detail above, I have major concerns that the past cyanobacterial community is expressed in full in the data set and additional measurements using the full range of known HGs are necessary to determine how and when the community of heterocystous cyanobacteria changed in the Baltic Sea.

p. 3, l. 35. Although it is an interesting idea, I do not really see the need and use of HGs as paleo-proxy to trace back anoxic events. There are other lithological and/or bulk-geochemical means that are better suited to investigate sediments for anoxic events. Also, do all anoxic events have to be characterized by the presence of HGs? I assume not as this depends on the nature of the bloom-forming cyanobacteria. Cyanobacterial HABs can also be caused by unicellular or filamentous non-heterocystous cyanobacteria and there is evidence that these cyanobacteria can be abundant in the Baltic Sea as well. Blooms of these types of cyanobacteria may also have occurred in the past

Baltic Sea, causing anoxia but no HGs would be produced and hence the anoxic event would not be visible in the HG downcore record.

p. 4, l. 12-13. Please check the description of the sampling resolution. How can samples from 0-377 cm be sampled as 1 cm slices and samples from 241-377 cm be collected simultaneously as 2 cm slices?

p. 4, l. 17-26. Some of the descriptions of how the bulk-geochemcial data have been obtained are not clear to me but they are essential to understand whether the data is robust or not. I find the description on how the TC, TIC and TOC content of the MUC sediments were obtained very confusing. Was the procedure identical to the measurement of the TOC content of the GC sediments? It is also described that stable carbon isotope values of organic matter were determined. Yet, no stable carbon isotope values are given in the manuscript? So, this does not have to be described here.

p. 5, l. 5. I am wondering why the reproducibility of HG measurements on the GC sediments is less robust?

p. 5, l. 8. What do the abbreviations HDI26 and HDI28 stand for? They should be explained. Also, some information on the temperature calibration should be provided. Have those been established for the Baltic Sea? Have they been tested in the Baltic Sea and are they applicable in this type of setting?

p. 5, l. 18. Please check the timing of the LIA again. I am fairly sure that this cold interval did not extend until the 1950s.

p. 5., l. 25-26: The phrasing suggests that only six C6 HGs could be detected in the Baltic sediments. Given the information in the 'materials and methods' section, however, these six C6 HGs were the only HGs for which the sediments were investigated. Again, this should be clearly expressed in the manuscript.

p. 6, l. 7-8. I am intrigued by the difference in HG abundance although the overlapping

sediment sequences should represent the same time interval. Can this be a result of different preservations at the different sampling sites?

p. 7, l. 25. HAB could indicate all kinds of harmful algal blooms and should be replaced by 'cyanobacterial HAB'

p. 7, l. 35. See above comment and replace 'harmful algal blooms' by 'cyanobacterial HABs'

p. 8, l. 31-32. Again, I do not think that this conclusion is a valid at this stage. The study is largely limited to HGs produced by cyanobacteria belonging to the Nostocaceae and of course then they will always appear as major bloom-former. The full suite of HGs should be analyzed to comprehensively reconstruct the past cyanobacterial community.

p. 9, l. 2-3. Do the authors have other evidence to proof this? For example, indications from bulk-geochemical data, such as increased sulphur content or biological markers specific for a more marine algae community? This would be important to determine whether changes in the HG distribution and thus cyanobacterial community are indeed caused by inflow of salt water or not.

p. 9, l. 27. In the 'materials and methods' section, SWT has been introduced as 'surface water temperature' and here it is referred to as 'sea water temperature'. Which term is correct? The latter implies that the proxy is applicable in marine systems? Is that the case?

p. 9, l. 28. Here it is indicated that the temperature equations are based on cultures. While checking the original literature, however, I noticed that the equations in the culture study by Bauersachs et al. (2014; OG) are different from the once reported here. It seems that the equations used are actually taken from Bauersachs et al. (2015; Biogeosciences) and refer to a lacustrine environment.

p. 11, l. 3. I find it very difficult to follow the authors here. Multiple times it is indicated

that periods of bottom water anoxia occurred and temperatures changed over the investigated sediment profile. It would be advantageous if the intervals characterized by bottom water anoxia would be clearly indicated in Figure 3. Also, is there a temperature reconstruction that can be shown together with the TOC and nitrogen isotope records? It is suggested that temperature changed at the AL-LS transition and throughout the LS but there is no evidence for this provided in the manuscript. It would be very helpful if such data would be shown.

p. 11, l. 9-10. Based on the declining abundance of HGs, it is concluded that cyanobacterial HABs may have been less common and intense in the past brackish Baltic Sea. Although this might be the case, I miss a more thorough discussion of the HG data in context with other studies that have reconstructed past cyanobacterial activity in the Baltic Sea and that contradict with the findings of this study, showing that cyanobacteria were apparently abundant at least during the initial LS phase. In fact, the decreasing stable nitrogen isotope values at the start of the Littorina Sea shown in this study suggest a higher contribution of diazotrophic biomass to the organic matter content even though HG do not increase in this time interval. This actually raises the question whether HGs experienced significant degradation or not. This is a very essential issue to discuss and investigate if any robust conclusion on the use of HGs to trace cyanobacterial HABs and communities over time shall be made. One way to address this issue would be to investigate for other biomarkers specific for cyanobacteria, such as methyl branched alkanes or pigments to determine if they show similar trends as HGs.

p.11, l. 21-22. It may not lead to a complete destruction of HGs but can the authors rule out any effect of selective degradation on HGs? For example that shorter chain HGs are more easily degraded than their longer chain homologues? If that is the case, does the sedimentary HG distribution allow for the reconstruction of past cyanobacterial communities?

Figure 3. I noticed that nitrogen isotope values only for the GC have been obtained.

This is unfortunate as no link between the intensity of N2 fixation and the abundance of HGs can be made. It would be interesting to compare whether nitrogen isotope values are low in the MUC surface sediments and coincide with high HG abundances. If that is the case, it could be estimated how much HGs should be present e.g. in the initial LS phase and compared with the measured HG abundance. There are certainly uncertainties with such a calculation but it may help to clarify whether HGs should have been present in deeper parts of the records or not.

Technical Corrections

l. 36. Change to: …...the fixation of N2

p. 3, l. 17-18. The authors should be more consistent in the choice of terms. On multiple instances in the manuscript different expressions for cyanobacterial blooms are used (e.g. 'cyanobacterial HAB', 'cyanobacterial bloom' or 'HAB'). The latter is particular confusing because it could mean any harmful algal bloom not specifically those caused by cyanobacteria. The authors should just stick to one abbreviation.

p. 3, l. 22. Change to: 'water column stratification'

p. 3, l. 32. Change to: 'anoxic events'

p. 3, l. 36-37. In the remainder of the text, 'total organic carbon' is used. Why not here as well?

p. 3, l. 37. Please change to 'nitrogen isotope record'

p. 3, l. 38: Change to 'specific biomarkers of'

p. 4, l. 3. Change to 'max. 248 m'

p. 4, l. 11. This should be '7200 cal. kyr BP', shouldn't it?

p. 4, l. 14. Delete 'of' in 'grounded before of further'

p. 5, l. 1. 'IPL' was not introduced before.

p. 6, l. 20-21. Change to 'C28 keto-diol HG'

p. 6, l. 27. Change to 'during the brackish phase'

p. 6, l. 31. Change to 'HG distribution'

p. 6, l. 32. Change to 'HG distribution'

p. 8, l. 24. Change to 'or occurred in traces'

p. 9, l. 37. Change to 'cyanobacterial HAB'

p. 12, l. 4. Change to 'summer cyanobacterial HABs'

Figure 3: The delta symbol is not displayed correctly.

---

## Author Response (AR1)

In this rebuttal, we respond to the comments made by the AE and referees by first repeating their statements (in italic font), followed by our reply (in regular font). We refer to page numbers and line numbers of the revised manuscript, unless otherwise stated.

**Reply to the comments by Associate Editor (Prof. Clare Woulds)**

*Thank you for your comprehensive responses to reviewer comments. I would like to invite you to submit a revised manuscript, containing the revisions that you have detailed, as well as your responses to the main points made by reviewers. As part of this I do not require the additional data suggested by Reviewer #3. Although this would certainly be interesting, I recognise that it is beyond the scope of the present study.*

We thank the AE for the timely decision. The paper has been adjusted along the lines discussed in the original responses posted on line earlier. In addition, we obtained some recent data that affected the stratigraphy of the cores; it turned out that the deepest core also contained sediments deposited during the Yoldia Sea phase. We have adjusted the text and the figures accordingly. We also added a new co-author as it turned out that she contributed substantially in obtaining the nitrogen isotope record. All co-authors have been informed about this change and agree with it. We hope that the current version is now acceptable for publication in Biogeosciences.

*In addition I would appreciate it if you could accommodate the following two comments of mine, from my initial reading:*

*The introduction does not explain why this additional proxy record should be expected to provide a more definitive answer as to the question of whether or not HABs and anoxia are linked to human activity.*

The general purpose of our work is not to check for the influence of human impact. We formulated the goal as follows: "In this study, we test the potential of HGs as paleo-proxy to investigate the changes in past communities involved in the summer cHABs in the Baltic Sea over the Holocene and the potential relationship with the anoxic events that occurred in the basin." (page 5 lines 1-3). In an earlier paragraph we mentioned the potential human impact on the occurrence of cHABs but we also reference studies that claim that cHABs occurred within the Baltic Sea before the impact of human mankind.

*Page 3 line 35 – states that HGs will be used to trace anoxic events in the past. However, they are not a proxy for anoxia, only for the presence of cyanobacteria. Please amend this wording (this may already have been done in response to reviewer comments).*

This comment has been accommodated in the revised version in the formulation of the goal of our work (page 5 lines 1-3; page 11 lines 4-7).

**Reply to the comments by Referee #1.**

*This paper reports on the downcore distribution of heterocyst glycolipids (HGs) in the Baltic Sea in an attempt to evaluate the utility of HGs as tracers for past nitrogen fixation by cyanobacteria. The rationale is very well formulated and the data are unique and precious given the high-resolution sedimentary record and the limited information on these biomarkers in paleo-environment studies. I am very impressed with the depth of analytical analysis involved and the motivation of research. However, I must admit that I am not convinced that the data deliver the conclusion described in the abstract. I have two major concerns.*

We thank referee #1 for taking the time to read and comment on our manuscript. We do appreciate the positive assessment of our work. Here we respond to the various issues raised in the two concerns of the referee.

Concern #1:

"*My first concern relates to the preservation of HGs in sediments (as is briefly discussed by the authors in the text as well). How does HG decomposition vary in freshwater versus brackish water systems? In modern freshwater and brackish water systems, does HG composition show the same pattern as observed in the sediment core? Is it possible that HGs are better preserved in brackish waters, leading to their higher abundance as well as stability compared to in freshwater systems? If so, HGs in sediments are not only related to their inputs but also to their decay. As both processes are influenced by temperature, the presence of O2 and possibly salinity, it is very difficult to conclude on "the potential of HGs as specific biomarker of heterocystous cyanobacteria in paleoenvironmental studies". Instead, I would suggest considering whether there is a proxy or indicator that may be used to (even roughly) assess the preservation or degradation stage of HGs in sediments? In lines 27-35 (pg 9), it is mentioned that sea surface temperatures reconstructed using HGs were too high to be realistic and the causes were not clarified. To me, this seems like a hint that HG signatures in the sediments may be subject to diagenesis-related alterations and that different molecules have be influenced differentially. I think the authors need to clarify this possibility before making conclusions and in the abstract as well.*"

This concern of the referee relates to two fundamental questions: what is the effect of changes in preservation conditions on 1) the concentration and 2) the distribution of HGs in sediments. Indeed the ability of HGs to be preserved in sediments represents a key premise to our work (as it is for every biomarker). We discuss potential break-down of HGs in sediments quite extensively (pages 10–11, lines 30 – 40 and 1 – 22 in the original manuscript) but this is mainly related to the breakdown in the anoxic sediments (i.e. the marked decline in the concentration of HGs in uppermost sediments). We have interpreted their decline in the sediments deposited during the freshwater phase as a much lower abundance of nitrogen-fixing cyanobacteria (which is supported by the increased $\delta^{15}N$ values) but it is true that we should have discussed more extensively the fact that the changing redox conditions in the surface sediments (i.e. from anoxic to oxic) will probably have affected the conditions for preservation of HGs. To compensate for this we normalized the HG concentrations on TOC but it is known that biomarkers are more readily degraded than TOC under oxic conditions. In the revised version of our manuscript we have discussed this topic more in-depth (page 18 lines 28-33 and page 19 lines 1-9). On the matter of how the HG decomposition varies in modern freshwater versus brackish water systems (i.e. effect of salinity), to the best of our knowledge, no studies are available. However, we think that other environmental parameters (e.g. oxygen exposure) are far more important.

The second issue that was raised concerns the question whether perhaps partial degradation of HGs results in significant changes in the distribution of the HGs in such a way that differences in HG distribution, interpreted as arising from a difference in the composition of heterocystous cyanobacteria (e.g. as done in our study for the differences observed between the Ancylus Lake and brackish Baltic Sea), is in fact caused by differences in degradation of individual HGs. We don't feel that this is likely. First of all, all HGs are chemically quite similar and we don't expect large differences in oxic degradation rates. Secondly, in a number of systems a good match in the distribution of the HG distribution of suspended water column and surface and (in some cases) deeper sediments is observed (i.e. Lake Challa, Bauersachs et al 2010; Lake Schreventeich, Bauersachs et al., 2015; equatorial Atlantic, Bale et al., 2017). Upon sedimentation, a substantial fraction of the HGs will be degraded, so these studies indicate no preferential degradation of specific HGs. Lastly, the HG distribution that we find in the surface sediments of the Baltic Sea are fully in line with the HG composition of the most important heterocystous cyanobacteria (see text of the manuscript; now expanded with the recent data of Bauersachs et al., 2017). Consequently, we don't see this as a problem and have full confidence in using the HG distribution to infer potential sources but we have elaborated on this a little bit more in the revised version of our manuscript (page 19 lines 6-9).

Concern #2:

*"My second concern relates to the influence of multiple environmental variables on HG composition and distributions. As the authors mentioned (several times) in the text, HG variations may be related to temperature variations as well as salinity changes. I think that control experiments are needed to prove that HG shifts are related to cyanobacteria community changes only instead of being affected by physiochemical processes also."*

We do agree with the suggestion that controlled experiments would help to further elucidate the influence of environmental factors such as temperature and/or salinity on the HG composition. As indicated by the referee, we already quote quite a number studies that have examined the influence of temperature and these studies have even resulted in the potential application of HGs to reconstruct temperature. Studies on the effect of salinity on the HG composition of different heterocystous cyanobacteria have not been performed but are more difficult because of the restricted salinity range of heterocystous cyanobacteria. However, these kind of studies fall outside the scope of the present work that describes the HG composition of the Holocene sedimentary record of the Baltic Sea.

*A minor point: I am not sure if Figure 6 provides any new information. This is just another form of Figure 4. The authors may consider removing this figure.*

We understand this concern of the referee but would like to keep this data in as it shows the results of the statistical data treatment, which is in good agreement with the stratigraphy of the core. We, therefore, combined Figs. 4 and 6 into one figure (Fig. 4 in the new version).

**Reply to the comments by Referee #2.**

*This manuscript is a substantial contribution to developing a molecular proxy for N2- fixation in the geological record, namely the diagnostic glycolipids indicative of heterocyst envelopes. The manuscript is very well written, structured, and points are well argued. The amount of analyses is staggering and I definitely support publication of the paper. An impressive set of analyses of these heterocyst glycolipids (HGs) in dated sediment cores from the Baltic Sea are the basis on which the authors explore some very interesting ideas: The modern Baltic Sea is known for massive blooms of two species of cyanobacteria and there is evidence from molecular and isotope data that they occurred during much of the Littorina Sea (LS) Stage and subsequent brackish phases. Whether N-fixation was a feature in the pre-Littorina lacustrine stages was unknown (at least to me). A first objective was to test if the HG patterns give evidence for alternating communities mirrored in HG distribution patterns. That appears*

*indeed to be the case. HG patterns have been remarkably stable over the last 7000 years or so, although HG abundances in the short core show a typical decrease in their contribution to total organic carbon (TOC) that suggests that they are more rapidly degraded than bulk TOC (Fig. 6 upper left). Below 200 cm in the longer core, both abundances relative to TOC and HG composition are highly variable. These variations characterize the lacustrine Ancylus Lake (AL) Stage, that must have received input of organic matter containing HGs.*

We thank referee #2 for the comments and suggestions.

*What are the HG patterns of soil cyanobacteria, and is the input of soil-derived TOC a possible source and also possibly a reason for differences in AL and LS sediments? I seem to remember that lignin biomarker abundance increased at the AL/LS transition.*

The referee is correct that $N_2$-fixing cyanobacteria occur in soil and some of them belong to the heterocystous cyanobacteria (e.g. *Nostoc* and *Calothrix* species) and may produce HGs. Erosion of soil could thus potentially lead to an influx of HGs in Baltic Sea sediments. However, since the HG lipids contain an attached sugar moiety, we feel it is unlikely that HGs produced in soil will make it to the sediments of the Baltic Sea since they would be exposed extensively to oxygen during transport and only relatively stable components such as lignin, wax lipids, and branched GDGTs will likely survive this transport to the middle of the Baltic Sea where our core was taken. It is also unknown whether HG-producing cyanobacteria occur in soils surrounding the Baltic Sea and if they are sufficiently abundant to account for the relatively high levels of HGs in the Baltic Sea sediments. We have discussed this briefly in the revised version of our manuscript (page 13 lines 24-30).

*What are the levels of r.u. compared to other depositional settings? Is the Baltic Sea particularly rich in HGs?*

It is, at the moment, rather difficult to answer this question. The mass spectrometric response for HGs is quite variable over time, which makes it difficult, if not impossible, to compare different data sets since samples have not been run during the same batch. We have run the set of Baltic Sea sediment samples in one batch and so we are confident that we can compare the HG concentrations relative to each other (i.e. the trend in Fig. 3a) but we feel we are not in a position to compare concentrations between different set of samples. To this end, we would need an HG standard that allows to monitor the mass spectrometric response over time. Such a standard has recently become available in our lab (Bale et al., Organic Geochemistry, 2017) and so this question may be answered in the future but falls outside the scope of the current study.

*A second (and interesting) objective was to investigate a fundamental biogeochemical feedback: Because the brackish Baltic Sea (LS and younger stages) experienced several alternations between oxic and anoxic conditions, it is a well-chosen environment to investigate whether or not development of anoxia and the Redfield homeostat (nitrogen fixation balancing a surplus of P originating from sediment or from denitrification) are linked, and if cyanobacterial biomass has an influence on the development of anoxia (or if anoxia had an influence of HG production). This is a difficult question and I wonder if it can be answered at all if you normalize your r.u. to %TOC. Are the unnormalised r.u. linearly correlated with %TOC? Figure 3a in comparison to 3e suggests this. That would mean that TOC preserved is the overriding control on HG abundances (but not composition) – by normalizing to TOC, any variation in HG abundance will then be masked. If TOC is high in anoxic and low in oxic phases, the effects of production and preservation can in my opinion not be segregated.*

The issue raised is a general one; i.e. do you normalize biomarker concentrations on g dry weight sediment or on TOC? To our opinion, if we are interested in reconstructing water column processes by examining biomarker profiles, like we are here, one should always compensate for the dilution of organic matter (and thus the biomarkers) by the inorganic matter of the sediment. To illustrate this: if we dilute a TOC-rich sediment with a tenfold amount of inorganic matter, the biomarker concentration normalized on g dry weight sediment will drop by a factor of 10, whereas that normalized on TOC will remain the same. Normalization on TOC will also compensate, to a certain extent, for different degrees of oxygen exposure (see also comments by referee #1), although biomarkers are generally more susceptible (i.e. higher degradation rates) than TOC, resulting in a decrease in concentration even when normalized on TOC. However, normalisation on g dry weight sediment will not solve this, so TOC-normalized biomarkers are to be preferred. This means that the TOC-normalized HG-concentration record provides the best possible insight in the presence of $N_2$-fixing cyanobacteria in the Baltic Sea. However, as extensively discussed in our manuscript, there are some issues (such as the rapid decline in the surface section) that prevent us making definitive conclusions with respect to the biogeochemical issues raised by the referee. We now briefly mention in the manuscript why we normalize on TOC (page 18 lines 28-33 and page 19 lines 1-9).

*Does the downcore decrease in PC1 in the MUC mean that the HG are more labile than bulk TOC? In particular, the relative abundances in Figure 4 suggest to me that C28 keto-diol and C26 keto-ol must be more rapidly degraded that the other moieties. Have you analysed the principal components for the MUC and GC separately, and are the score patterns similar to those for the entire sample pool?*

As discussed in the manuscript, we interpret the downcore decrease in the TOC-normalized HG-concentration in the MUC as either degradation, indicating that they are more labile than bulk TOC, or a decrease in the yearly occurrence of cyanobacterial blooms. However, the potential higher lability of the HGs has nothing to do with the declining scores on PC1 in the MUC since the principal components analysis was performed on the HG distributions. This decline can indeed be attributed to a decrease in the fractional abundance of the C28 keto-diol and C26 keto-ol as shown in Fig. 3. This could be due to preferential degradation of these HGs as suggested by the referee but we interpret this as a subtle change in the composition of the HGs produced in the water column. Indeed, we feel that it is unlikely that there will be a major difference in the degradation rate between the various HGs since their structures are quite similar. We have now mentioned this in the revised version of our manuscript (page 19 lines 6-9). We did perform principal component analysis on the distributions of the HGs separately but this does not reveal substantial changes. We do feel it is more logical to perform it on the complete set since all sediments comprise the Holocene sedimentary record of the Baltic and separating them in two is just artificially depending on the sampling techniques.

*Why do some labs continue to use acidified samples for d15N analyses in the face of ample evidence that this affects the values? But that is not crucial to this paper.*

The referee is correct that there is evidence for alteration of the $\delta^{15}N$ values when samples are acidified. This problem is especially evident in ecological studies, where $\delta^{15}N$ of biomass (which contains a lot of labile organic nitrogen) is present. In sedimentary organic matter most of the really labile nitrogen has been removed already but there are indeed studies that show effects of acidification, so this is a fair comment. We are aware of this problem and we checked the experimental procedure (which was performed by a technician in our lab) and it turns out that we did not perform the $\delta^{15}N$ analysis on the decarbonated sediment samples but on separate non-treated samples. So, the description of our experimental procedure is not accurate and we have adjusted it in the revised version of our manuscript (page 5 lines 25-31 and page 6 lines 1-4). We thank the referee for spotting this since it has raised some concern (see also the comment of Dan Conley).

We have also addressed all minor issues raised by referee #2.

**Reply to the comments by Referee #3.**

*The manuscript reports on the use of lipids specific to heterocystous cyanobacteria (heterocyst glycolipids) as (i) tracers for investigating past changes in the community of cyanobacterial blooms and (ii) paleo-proxy to trace back anoxic events in the Holocene Baltic Sea. Sediments sampled from a multicore and a gravity core collected in the Gotland Basin have been investigated for bulk geochemistry, nitrogen isotopes and the distribution and abundance of heterocyst glycolipids. While the use of heterocyst glycolipids as biomarkers to trace for cyanobacterial blooms in the Baltic Sea is principally interesting, I have some major concerns regarding the experimental setup, study design as well as data acquisition and interpretation that have to be addressed before I can recommend publication of the manuscript.*

We thank the referee for the extensive review which will undoubtedly improve the quality of our manuscript. We regret that the referee does not really appreciate the advances that are reported in this manuscript, where we, for the first time, report a Holocene high resolution HG record based on >250 samples. The referee also makes an unrealistic request for additional data. In summary the referee requests:

1) Re-extraction of all 250 samples with the more "gentle" Bligh-Dyer method
2) Reanalysis of the whole dataset of 250 samples with a method able to detect the "full range" of HGs
3) Reanalysis of the whole dataset in a quantitative fashion using an internal standard instead of the semi-quantitative method used here
4) Generation of an independent temperature record based on biomarkers
5) Examination in all 250 samples of the distribution of branched hydrocarbons
6) Examination of HG distributions of heterocystous cyanobacteria isolated from the Baltic Sea
7) Generation of $\delta^{15}N$ data for the multicore (100 samples)

This "shopping list" of the referee, which probably would require at least one full year work in the lab, is highly unrealistic and strongly contrasts the assessments of referee#2, who says that "the amount of analyses is staggering".

*My first and most pressing concern is related to the reconstruction of the past Baltic Sea cyanobacterial community, which seems to be the major aims of the study. From reading the manuscript, I got the impression that only six C6 HGs were present in the Baltic Sea sediments and that those are indicative mainly for heterocystous cyanobacteria of the family Nostocaceae; in agreement with the major bloom-forming Baltic Sea cyanobacteria. However, while having a closer look at the method used for the detection of HGs (apparently the same method described by Bale et al. (2015; OG)), I could not fail to notice that the method specifically targets only these six C6 HG but it is neither able to detect HGs of longer chain length (e.g. C30 to C32 keto-ol, keto-diol, diol and triol HGs), which have been described from numerous heterocystous cyanobacteria previously (Gambacorta et al. 1998; Phytochemistry) nor HGs with deoxyhexose or pentose headgroup (attached to a C26 alkyl chain) as described by Wörmer et al. (2012; L&O). This essentially means that the authors limit themselves to a very narrow window of HGs and consequently members of the cyanobacterial community that can be detected with their method. Moreover, they limit*

*themselves largely to the detection of Nostocaceae. So, my major concern is: do the presented HG profiles really reflect the complete cyanobacterial community or in fact only a small fraction of the community and is it then possible to draw any conclusion on the cyanobacterial community at all? It is very much likely that HGs with higher carbon chain length or other sugar head groups are also abundant and perhaps also more dominant than the six C6 HGs that were detected in the Baltic Sea sediments but we would never know because they are not included in the detection method. This might be in particular the case for the freshwater interval, for which major changes in the cyanobacterial community would be expected.*

The referee is correct that the method used targets the six HGs specified. We have now more clearly stated this throughout the manuscript. This choice is based on earlier analyses of Baltic Sea surface sediments (Bauersachs et al., 2010), published work on the composition of HGs in heterocystous cyanobacteria, and on microbiological studies of the occurrence of cyanobacteria in the Baltic Sea. A recent extensive meta-omics study revealed that in the Baltic proper (the predominant area for cyanoHABs) 69% of the heterocystous cyanobacteria belong Aphanizomenon, 23% to Dolichospermum (formerly Anabaena), and 8% to Nodularia. Bauersachs et al. (2017) have recently analysed HGs of eight representative heterocystous cyanobacterial strains isolated from the Baltic Sea and the six HGs analysed in our study form by far the majority (i.e. 97.7-100%) of the HGs. HGs with longer alkyl chains were not detected. The study concludes (last sentence of the abstract) "As heterocystous cyanobacteria of the genera Aphanizomenon, Dolichospermum and Nodularia are generally known to form massive blooms in many brackish as well as lacustrine systems worldwide, the chemotaxonomic markers introduced in this study may allow investigating cyanoHABs in a great variety of contemporary environments from polar to tropical latitudes". So our choice of HGs is very much in line with this independent study. Furthermore, we have analyzed a selected sample set from the brackish period with full scan HPLC-orbitrap-MS, allowing to make an inventory of all the HGs present, and were not able to find C28+ HGs. These results have now been added to the manuscript (page 8 lines 27-30).

The referee is right that perhaps the HG distribution during the Ancylus lake phase may have comprised HGs outside our analytical window (i.e. the six HGs) because conditions were quite different from those today. We have mentioned this in the revised manuscript and have adjusted the discussion to some extent (page 13 lines 15-30). However, we still feel that our data clearly demonstrates that a substantial change in the heterocystous cyanobacterial community must have occurred. This is carefully formulated in the abstract of the manuscript (p. 1, l. 26-28 of the initial submission); "During the earlier freshwater phase of the Baltic (i.e. the Ancylus Lake phase) the distribution of the HGs varied much more than in the subsequent brackish phase and the absolute abundance of HGs was much lower than during the brackish phase."

The referee suggests that all 250 samples should be reanalyzed for HGs before this study can be accepted. This is not possible since these analyses have been performed almost four years ago and we have noted that after filtration of extracts degradation of HGs may occur. Furthermore, for proper quantification, samples have been reanalyzed several times in that period and, especially for the samples with much lower amounts of HGs (i.e. the Ancylus lake phase sediments) there is certainly no material available anymore. We strongly feel that this suggestion for reanalysis is unjustified. Although our study has some limitations (as probably does every scientific study), the results provide novel insight into the use of HGs as potential markers for past blooms of cyanoHABs and is very timely.

*Moreover, it makes of course sense that the presented HG profiles agree with the major bloom-forming genera if only those HGs are included in the detection method that are*

*specific for cyanobacteria of the family Nostocaceae. In my opinion, the authors may have missed major changes in the cyanobacterial community due to the limited number of HGs that have been investigated. To obtain robust results and hence make reliable interpretations of cyanobacterial community changes over time, most if not all of the samples would need to be re-measured using a method that includes the full spectrum of HGs currently known from heterocystous cyanobacteria. This identification of potential biological sources of HGs in the Baltic Sea sediments is similarly problematic. I got again the impression that the six C6 HGs shown in Table 2 and discussed in the text cover the full spectrum of HGs that are present in the listed heterocystous cyanobacteria. From reading the original literature, however, it seems that many of these species do not only contain the six C6 HGs but also other HGs in substantial abundances, in particular when they do not belong to the Nostocaceae. For example, according to the authors, Tolypothrix contains only C28 diol and keto-ol HGs but in fact it also contains significant quantities of C30 triol and keto-diol HGs that surprisingly have not been included in the table and again this component will be missed in the Baltic Sea sediments as it is not included in the detection method. Likewise, according to the authors the C28 triol HG should be the only HG present in Scytonema hofmanni. This by no means is the case if the original literature is consulted (Gambacorta et al. (1998; Phytochemistry)). In fact, this HG is not present in S. hofmanni at all. Instead, it only contains C30 triol and keto-diol HGs, both of which cannot be detected using the analytical protocol described in the present study. There are other examples, such as Aphanizomenon aphanizomenoides or Aphanizomenon ovalisporum, from which only incomplete HG profiles are described lacking e.g. C30 diol, triol, keto-ol and keto-diol HGs as well as HGs with deoxyhexose or pentose head group. I am wondering why only a selection of HGs is shown in the table and why this is biased towards Nostocaceae? In any case, the question remains: Can the cyanobacterial community reconstructed based on these incomplete records? I think not. Therefore, I strongly encourage the authors to carefully check the table and where necessary to complete the full range of HGs. Otherwise, it is not possible to link HG profiles detected in the sediment record to the biological sources of HG and any attempt to reconstruct cyanobacterial community changes will be flawed.*

This repeats much of what has been said before. We have extended Table 1 with the new HG data of Baltic Sea cyanobacterial strains (Bauersachs et al., 2017), checked some of the issues mentioned by the referee, and more clearly indicate that this table lists the occurrence of the six targeted HGs in cultures (see the revised Table 1).

*Although I generally appreciate the authors' efforts to identify the sources of HGs in Baltic Sea sediments, I have my doubts that this is possible by comparing sedimentary HG profiles with HG distributions in cultured cyanobacteria. All cyanobacteria investigated for their HG content so far include either freshwater or marine representatives but brackish species (such as those from the Baltic Sea) have not been analyzed so far. Given that the environmental conditions in the Baltic Sea significantly differ from freshwater and marine environments, it is likely that Nodularia, Aphanizomenon or Anabaena species living in the brackish Baltic Sea will not necessarily show similar HG profiles as found in freshwater and marine cyanobacteria. Although it requires additional work, the authors may consider including modern Baltic Sea cyanobacteria in their study, so that HG distributions can be unequivocally linked to their biological sources and eventually be used to reconstruct changes in the cyanobacterial community in the Holocene Baltic Sea. This would significantly strengthen their conclusions.*

This comment is no longer relevant now that Bauersachs et al. (2017) have published their work on the HG distribution of eights strains isolated from the Baltic Sea. We also note that some of the older HG distributional studies are based on some strains isolated from the water colum of the Baltic Sea (i.e. Nodularia sp., CCY9414 and CCY9416; Aphanizomenon sp. CCY9905). We are surprised to learn that this referee doubts the usefulness of culture studies for biomarker studies since this basically forms the basis for the interpretation of the fossil biomarker record.

*While having a look at the HG structures, I am wondering if accelerated solvent extraction is the method of choice for extracting HGs from sediments? For my feeling, this particular extraction method is too harsh and may lead to the degradation of HGs. I assume that there is a reason why other studies dealing with HGs (such as Bale et al. 2015 (OG), 2016 (L&O); Schouten et al. 2013 (Phytochemistry); Bauersachs et al. 2009 (Phytochemistry); 2015 (Biogeosciences)) have used the more gentle Bligh and Dyer extraction method? Can the authors proof beyond doubt that the extraction method did not flaw the generated HG profiles and that these profiles are indeed representatives for the sedimentary signal? While reading some of the original literature, I noticed that a comparison between ASE and Bligh & Dyer extraction has been made previously (Bauersachs et al. 2010; PNAS). Yet, the comparison is only semiquantitative and without quantification using a standard also less robust. From these experiments it is also not clear whether ASE leads to selective degradation of keto-ol vs diol HGs or diol vs triol HGs. This is likely not an issue with Bligh and Dyer but with ASE it may indeed be problematic. This issue, however, it is not addressed in the manuscript. As it is now possible to quantify HGs (see Bale et al. 2016; OG), these experiments could easily be done and included in the manuscript.*

The referee has seen already that we have tested three extraction methods for HG determination, which showed that slightly higher yields were obtained by the ASE extraction used in this study. This study also clearly showed that the three extraction methods yielded almost identical HG distributions. The recently published method which uses an internal standard (see Bale et al. 2017; OG) became only available two years after all analyses were completed. As explained it is not possible to reliably re-analyse these samples. Furthermore, we have taken great care in the semi-quantitative analysis of the HGs and have duplicated all (i.e. 250 samples) our analyses, which is commonly not done in such studies. The request for "proving beyond doubt that the extraction method did not flaw the generated HG profiles and that these profiles are indeed representatives for the sedimentary signal" can basically be made for every study working on organic components in sediments and follows the general strategy of this referee to ask for more work to be done wherever possible. Doubt is a highly personal and non-scientific expression in this sense.

*The issue of how degradation may affect HGs is also only little addressed but essential to verify the robustness of these components as biomarkers for cyanobacterial HABs in the Baltic Sea and as paleo-proxies. As stressed by the other reviewers, the HG profile does not really match other profiles of cyanobacterial activity such as those based on cyanobacterial pigments reported by Bianchi et al. (2000; L&O) or Funkey et al. (2014; EST). Yet, the nitrogen isotope record for instance shows lowest values at the AL-LS transition and the lower part of the Littorina Sea phase, which may point to an increased loading of nitrogen derived from cyanobacterial N2 fixation. Therefore, it is surprising that this interval is not characterized by increased abundances of HGs. As indicated in the text, heterocystous cyanobacteria may not have formed blooms in the past Baltic Sea but this is in contrast to previous findings and certainly needs more attention in the manuscript. It could also very well be that HGs experienced some sort of degradation and are therefore not abundant in the lower part of the Littorina Sea phase anymore. Determining the degradation of HGs is certainly beyond the scope of the manuscript but it would be interesting to obtain additional proof for the presence/absence of cyanobacteria along the record. One such proxy is pigments but distributions of methyl branched alkanes (a well-established marker for*

*cyanobacteria) are an alternative. If these independent proxies show similar profiles as the HGs, this would at least strengthen the authors' hypothesis that cyanobacterial HABs played only a minor role in past Baltic Sea.*

This referee clearly studied the earlier reviews of our manuscript in quite some detail. We responded to these concerns in our responses (both on the degradation issue and the use of carotenoid pigments) and do not repeat these arguments here. This referee also suggests that we should check our large sample set (>250 samples) for other biomarkers, i.e. branched alkanes. Branched alkanes are certainly not specific enough as biomarkers for heterocystous cyanobacteria as there are also reports of these components in other non-heterocystous cyanobacteria and algae. We also would like to remind the referee that we concluded from our work that "the abundance of HGs dropped substantially with depth and this may be caused by either a decrease of the cyanobacterial blooms or diagenesis, resulting in partial destruction of the HGs" (p. 1, l. 22-23) so we do not directly suggest that cyanobacterial HABs played a minor role in the past Baltic Sea".

*The use of the temperature indices does not seem to add much to the manuscript and I am wondering if it is really needed. Temperatures reconstructed using the HDI26 or HDI28 are not described in the results section and only briefly touched on in the discussion section (p. 9, l. 20-35). I also find this discussion hard to follow. It is not really clear to me how they calculated the temperatures. It is indicated that proxy calibrations from cultures were used but the calibrations described in the 'materials and methods' section seem to be those established by Bauersachs et al. (2015; Biogeosciences) for a lake environment. This is confusing and it should be clarified which calibration has been applied before any robust discussion of temperature can actually be made. I also have my doubts that the culture or the lake calibrations are indeed applicable in the brackish Baltic Sea and that the 'somewhat unrealistic' and too high temperatures result from the lack of a calibration specifically established for the Baltic Sea. There are numerous other examples where specific calibrations have been generated for the Baltic Sea including e.g. the TEXL86. The publication would greatly benefit from including a temperature record based on a well-established temperature proxy, such e.g. the TEXL86. As mentioned on several occasions in the manuscript, 16 °C seems to be a sort of threshold with temperatures >16 °C promoting bloom-formation. If established and plotted along with the HG data, it would allow identifying intervals during which past cyanobacterial HABS may have occurred in the Baltic Sea.*

Temperatures are not reported in the results because they are NOT a result but an interpretation of the data and, therefore, belong in the discussion. We feel it is interesting to see if the HG distribution, as seen in lakes, may be an indicator of water temperature and we feel we should spend a few lines on this topic. Clearly, this referee wants to have some more details and we have expanded this section a bit (page 14 lines 13-32 and page 15 lines 1-6) and have included a TEX$_{86}$ data from this (Warden et al., 2017) and a related core (Kabel et al., 2012) in Fig. 4.

*Specific Corrections*

We appreciate that the referee has taken so much time to go in detail through the manuscript. The term "correction", however, seems to indicate that all items listed below deal with errors in our manuscript, which we feel is unsubstantiated.

*p. 1, l. 17-18. Please mention the different genera of bloom-forming heterocystous cyanobacteria.*

This has been added (page 1 line 21).

*p. 2, l. 7. Adams was certainly not the first one to describe the role of the heterocyst in the process of N2 fixation. A very nice overview on this topic is provided by Wolk (1982) and this author certainly deserves credit for his work. Please add the following reference to the manuscript: Wolk, CP (1982). Heterocysts. In: Carr, N.G., Whitton, B.A. (Eds), The Biology of Cyanobacteria. Blackwell Scientific Publishers, Oxford, pp. 359-386.*

We appreciate that the referee is concerned with the fact that original literature generally should be cited. However, we feel that this does not apply to review papers which provide a current overview on a topic. But it can't harm to add an older overview too, so we have added this reference (page 2 line 22).

*p. 2, l. 7-9. The way the sentence is phrased, it seems that both the polysaccharide and the glycolipid layer are involved in regulating the diffusion of atmospheric gases to the heterocyst. Yet, the polysaccharide layer is considered to provide protection with regard to mechanical damages. Please rephrase to make clear that the glycolipid layer is the gas diffusion barrier.*

We have rephrased this sentence (page 2 lines 22-24).

*p. 2, l. 15: I do not like the term 'free-living' too much. I think 'non-symbiotic' is more appropriate in this context.*

We have changed this according to the referee's suggestion (page 2 line 32).

*p. 2, l. 31. A reference to studies addressing the nature of cyanobacterial HABs is missing here.*

We have added a reference (page 2 lines 12-13).

*p. 3, l. 8. In addition, species of the genus Anabaena may also be important bloom formers and they should be included here. They are mentioned as bloom-formers in the discussion section. So why not here as well?*

Point taken; we now also refer to the recently published detailed meta-omic analyses of Celepli et al.(Environm. Microbiol., 2017) (page 4 lines 5-8).

*p. 3, l. 33. The authors state to investigate 'past cyanobacterial communities' but in fact they limit themselves to a very narrow range of the cyanobacterial community as their method only allows the detection of six C6 heterocyst glycolipids. As expressed in detail above, I have major concerns that the past cyanobacterial community is expressed in full in the data set and additional measurements using the full range of known HGs are necessary to determine how and when the community of heterocystous cyanobacteria changed in the Baltic Sea.*

This has been extensively discussed earlier and we refer to our response.

*p. 3, l. 35. Although it is an interesting idea, I do not really see the need and use of HGs as paleo-proxy to trace back anoxic events. There are other lithological and/or bulk geochemical means that are better suited to investigate sediments for anoxic events. Also, do all anoxic events have to be characterized by the presence of HGs? I assume not as this depends on the nature of the bloom-forming cyanobacteria. Cyanobacterial HABs can also be caused by unicellular or filamentous non-heterocystous cyanobacteria and there is evidence that these cyanobacteria can be abundant in the Baltic Sea as well. Blooms of these types of cyanobacteria may also have occurred in the past. Baltic Sea, causing anoxia but no HGs would be produced and hence the anoxic event would not be visible in the HG downcore record.*

We are not claiming that HGs are the best proxies for tracing anoxic events. In this paper we are evaluating the use of HGs as potential proxies for CyanoHABs in the past since in the present day Baltic Sea since these HABs are of great importance in the development of anoxia. We do not exclude the possibility that other circumstances may also lead to anoxia. We agree that this sentence needs slight rephrasing and have adjusted this (page 5 lines 1-3).

*p. 4, l. 12-13. Please check the description of the sampling resolution. How can samples from 0-377 cm be sampled as 1 cm slices and samples from 241-377 cm be collected simultaneously as 2 cm slices?*

Well spotted. The section 0-241 cm was sampled as 1 cm slices. We have corrected this.

*p. 4, l. 17-26. Some of the descriptions of how the bulk-geochemcial data have been obtained are not clear to me but they are essential to understand whether the data is robust or not. I find the description on how the TC, TIC and TOC content of the MUC sediments were obtained very confusing. Was the procedure identical to the measurement of the TOC content of the GC sediments? It is also described that stable carbon isotope values of organic matter were determined. Yet, no stable carbon isotope values are given in the manuscript? So, this does not have to be described here.*

We have revised this section to make it more clear and left out the description of the stable carbon isotope analysis.

*p. 5, l. 5. I am wondering why the reproducibility of HG measurements on the GC sediments is less robust?*

So do we; it may relate to the overall lower HG concentrations in this section.

*p. 5, l. 8. What do the abbreviations HDI26 and HDI28 stand for? They should be explained. Also, some information on the temperature calibration should be provided. Have those been established for the Baltic Sea? Have they been tested in the Baltic Sea and are they applicable in this type of setting?*

We have expanded this description. As described and discussed in the Results and Discussion section these have been established for lakes. Our dataset just allows to test if they are applicable in the Baltic.

*p. 5, l. 18. Please check the timing of the LIA again. I am fairly sure that this cold interval did not extend until the 1950s.*

We thank the referee for noting this. We have checked the complete stratigraphy of the cores and made new age assignments, also based on new insights. The text and figures have been changed accordingly.

*p. 5., l. 25-26: The phrasing suggests that only six C6 HGs could be detected in the Baltic sediments. Given the information in the 'materials and methods' section, however, these six C6 HGs were the only HGs for which the sediments were investigated. Again, this should be clearly expressed in the manuscript.*

As described above, we have expanded this description to make this clear and have added the full-scan HPLC-orbitrap-LC experiments (the new Table 2).

*p. 6, l. 7-8. I am intrigued by the difference in HG abundance although the overlapping sediment sequences should represent the same time interval. Can this be a result of different preservations at the different sampling sites?*

We have discussed this in the manuscript and there is nothing we can add at the moment.

*p. 7, l. 25. HAB could indicate all kinds of harmful algal blooms and should be replaced by 'cyanobacterial HAB'*

*p. 7, l. 35. See above comment and replace 'harmful algal blooms' by 'cyanobacterial HABs'*

We have checked carefully the manuscript for this terminology and, where needed, adjusted it.

*p. 8, l. 31-32. Again, I do not think that this conclusion is a valid at this stage. The study is largely limited to HGs produced by cyanobacteria belonging to the Nostocaceae and of course then they will always appear as major bloom-former. The full suite of HGs should be analyzed to comprehensively reconstruct the past cyanobacterial community.*

How does this referee define "the full set of HGs"? When analyzing sediments one never can be sure that we look for all HGs that were once biosynthesized by heterocystous cyanobacteria residing the basin at that time. We have explained why the analysis was targeted on the six HGs analyzed in all the samples and we pointed out that in a suite of selected samples analyzed for all HGs these were by far the dominant HGs. Figure 4 clearly reveals that monitoring these 6 HGs allows to see changes in the composition of the heterocystous cyanobacteria over time and the relatively stable HG distribution does, in our view, allow us to say: "which suggests that the cyanobacterial community of the Baltic did not undergo major changes from the AL-LS transition to the MoWP and remained dominated by cyanobacteria belonging to the family Nostocaceae" (p. 8, l. 31-32 in the initial manuscript). We, as the authors of this manuscript, take the responsibility for this statement. It is an interpretation of the data set and the referee is free to disagree on this but cannot prescribe what we should do and should have done.

*p. 9, l. 2-3. Do the authors have other evidence to proof this? For example, indications from bulk-geochemical data, such as increased sulphur content or biological markers specific for a more marine algae community? This would be important to determine whether changes in the HG distribution and thus cyanobacterial community are indeed caused by inflow of salt water or not.*

In our revised chronology this period relates to the Ancyluc Lake phase II period,

*p. 9, l. 27. In the 'materials and methods' section, SWT has been introduced as 'surface water temperature' and here it is referred to as 'sea water temperature'. Which term is correct? The latter implies that the proxy is applicable in marine systems? Is that the case?*

We have fixed this.

*p. 9, l. 28. Here it is indicated that the temperature equations are based on cultures. While checking the original literature, however, I noticed that the equations in the culture study by Bauersachs et al. (2014; OG) are different from the once reported here. It seems that the equations used are actually taken from Bauersachs et al. (2015; Biogeosciences) and refer to a lacustrine environment.*

This issue was brought up earlier by the referee and we refer to our earlier response.

*p. 11, l. 3. I find it very difficult to follow the authors here. Multiple times it is indicated that periods of bottom water anoxia occurred and temperatures changed over the investigated sediment profile. It would be advantageous if the intervals characterized by bottom water anoxia would be clearly indicated in Figure 3. Also, is there a temperature reconstruction that can be shown together with the TOC and nitrogen isotope records? It is suggested that temperature changed at the AL-LS transition and throughout the LS but there is no evidence for this provided in the manuscript. It would be very helpful if such data would be shown.*

The intervals characterized by bottom water anoxia are clearly indicated in Figure 3 by the periods of deposition of laminated sediments. This is now clearly mentioned in the figure legend. We have also included in Fig. 3 the SST data of Warden et al. (Sci. Rep., 2017; data for the period 3000-7200 cal yr BP obtained for the same core) and data of Kabel et al. (2012) for recent palaeo SST data from basically the same site and discuss these data in the text where appropriate.

*p. 11, l. 9-10. Based on the declining abundance of HGs, it is concluded that cyanobacterial HABs may have been less common and intense in the past brackish Baltic Sea. Although this might be the case, I miss a more thorough discussion of the HG data in context with other studies that have reconstructed past cyanobacterial activity in the Baltic Sea and that contradict with the findings of this study, showing that cyanobacteria were apparently abundant at least during the initial LS phase. In fact, the decreasing stable nitrogen isotope values at the start of the Littorina Sea shown in this study suggest a higher contribution of diazotrophic biomass to the organic matter content even though HG do not increase in this time interval. This actually raises the question whether HGs experienced significant degradation or not. This is a very essential issue to discuss and investigate if any robust conclusion on the use of HGs to trace cyanobacterial HABs and communities over time shall be made. One way to address this issue would be to investigate for other biomarkers specific for cyanobacteria, such as methyl branched alkanes or pigments to determine if they show similar trends as HGs.*

This is a repetition of a concern raised earlier by this referee and by Dan Conley and we refer to our earlier responses.

*p.11, l. 21-22. It may not lead to a complete destruction of HGs but can the authors rule out any effect of selective degradation on HGs? For example that shorter chain HGs are more easily degraded than their longer chain homologues? If that is the case, does the sedimentary HG distribution allow for the reconstruction of past cyanobacterial communities?*

There have not been any studies on the effect of partial oxidation on HG distribution. Microbial oxidation is likely to start with hydrolysis of the glyosidic bond of the HGs. If so, this would mean that this has no substantial effect on the HG distribution since all HGs contain glycosidic bonds but this remains to be tested.

*Figure 3. I noticed that nitrogen isotope values only for the GC have been obtained. This is unfortunate as no link between the intensity of N2 fixation and the abundance of HGs can be made. It would be interesting to compare whether nitrogen isotope values are low in the MUC surface sediments and coincide with high HG abundances. If that is the case, it could be estimated how much HGs should be present e.g. in the initial LS phase and compared with the measured HG abundance. There are certainly uncertainties with such a calculation but it may help to clarify whether HGs should have been present in deeper parts of the records or not.*

The referee has much faith in $\delta^{15}N$ as indicator of nitrogen fixation. Initially, we thought we also obtained a $\delta^{15}N$ record for the MUC but this is not the case. Although, the proposed comparison of the HG and $\delta^{15}N$ record of the MUC proposed by the referee would be of interest, this would require substantial additional work which is beyond the scope of the present manuscript.

We have taken all minor technical "corrections" into consideration.

**Reply to the comments by Prof. Dan Conley**

We thank Dan Conley for his comments and provide answers to his two points as indicated below.

*1) It appears that the 15N samples were acidified before measurement, which has been shown in the literature to result in anomalous values. If you compare their 15N data with other data from the Baltic Sea from a variety of groups their data show very little variation through time especially during hypoxic periods.*

Please see our reply to a comment of referee #2; we did perform the $\delta^{15}N$ on untreated samples. The description of the experimental methods was inaccurate and has been adjusted. We apologize for this inconvenience. In contrast to what is mentioned by Conley, the range in $\delta^{15}N$ values we have measured in our core is slightly larger (1.2-5.2 permil) than that (2.0-4.5 permil) observed in the Funkey et al. (2014) study. Conley is also not right that the variation in the sediment in the main hypoxic phase during the Littorina transgression is less; it is comparable in magnitude (variations over slightly more than one permil). As requested, we have re-plotted the $\delta^{15}N$ data (the new Fig. 3b).

*2) The HG data also show a different picture than what has been observed with pigment biomarkers for cyanobacteria in the Baltic Sea. Funkey et al. (2014) – is referenced, but not discussed - showed increased cyanobacteria abundance during period of hypoxia likely due to changes in the biogeochemistry of P during low oxygen periods. I think more needs to be done to assure the validity of the 15N measurements and other proxies should be measured and compared to validate the HG data.*

Conley touches here on a sensitive problem. Yes, we did reference his and his co-workers paper but we did not discuss it extensively for two major reasons:

i) The carotenoids used in their paper, zeaxanthin and echinenone, are not entirely specific for cyanobacteria. Zeaxanthin commonly occurs in various classes of algae and higher plants; echinenone has a more limited occurrence but has been reported in bacteria and marine animals. These carotenoids are certainly not limited to nitrogen-fixing cyanobacteria, as opposed to the highly specific HGs that we use.

ii) As pointed out by referee #1, diagenesis (especially post-depositional oxidation) in this environment of highly variable sediment redox conditions should be considered when the sedimentary biomarker record is interpreted. Carotenoids are amongst the most unstable organic biomarkers because of their very labile conjugated system of double bonds. Changes in redox conditions will thus have a major effect on the concentration of carotenoids and hence interpretation of their concentration profile as a direct indication of the abundance of cyanobacterial nitrogen fixation is, at least in our opinion, somewhat simplistic.

Therefore, we don't really see any reason why the HGs should reveal a similar distribution to the much less specific and diagenetically more sensitive carotenoids. In view of this, we don't think we should measure these carotenoids as additional proxies. Our manuscript deals with the assessment of HGs as potential proxies for past cyanobacterial nitrogen fixation and, as indicated by both referee #1 and 2, provides an extensive study that does not require additional data.

Nevertheless, we now discuss the discrepancy with the, in our view, less specific carotenoid data published earlier (page 17, lines 20-32).

**References used in our response**

Bale, N., de Vries, S., Hopmans, E. C., Sinninghe Damsté, J. S., & Schouten, S. (2017). A method for quantifying heterocyst glycolipids in biomass and sediments. *Organic Geochemistry*, *110*, 33-35.

[revised manuscript text omitted]

| *Aph. aphanizomenoides* [b,c] F | UAM 523 | + | - | ++ | ++ | +tr | +- |
| *Aph. gracile* [b,c] F | UAM 521 | ++ | ++ | -tr. | - | tr. | - |
| *Aph. ovalisporum* [b,c,f] F | UAM 290 | ++ | +tr. | +tr. | +tr. | - | - |
| *Anabaena* sp. [a] | CCY 0017, 9910, | ++ | + | + | + | - | - |
| *Anabaena* sp. [a] | CCY 9402 | - | - | ++ | + | - | - |
| *Anabaena* sp. [a] | CCY 9613 | + | + | - | - | - | - |
| *Anabaena* sp. [a] | CCY 9614, 9922 | ++ | + | - | - | - | - |
| *Anabaena* sp. [b] | 315 | ++ | ++ | tr. | tr. | tr. | - |
| *Anabaena* sp. [b] | BIR53 | ++ | ++ | tr. | tr. | tr. | - |
| *Anabaena* sp. [b] | BIR169 | ++ | + | tr. | tr. | ++ | + |
| *Anabaena cylindrica* [a] F | CCY 9921 | ++ | + | - | - | - | - |
| *Anabaenopsis* sp. [a] | CCY 0520 | ++ | + | + | - | - | - |
| *Nostoc* sp. [a] | CCY 0012, 9926 | ++ | + | - | - | - | - |
| *Nostoc* sp. [b,c] | MA 4 | ++ | ++ | -tr. | - | tr. | - |
| *Cylindrospermopsis raciborskii* [b,c,f] F | UAM 520 | ++ | tr. | + | tr. | + | - |
| *Cyanospira rippkae* [d,e] F | ATCC 43194 | - | - | ++ | + | - | - |
| **Rivulariaceae cultures** | | | | | | | |
| *Calothrix desertica* [e,d] F | PCC 7102 | - | - | - | - | +
+ | ++ |
| *Calothrix* sp. [b,c] | MU 27 | - | - | tr. | - | ++ | ++ |
| *Calothrix* sp. [a] | CCY 0018 | - | - | - | - | ++ | +- |
| *Calothrix* sp. [a] | CCY 0202 | tr. | tr. | - | - | ++ | -+ |
| *Calothrix* sp. [a] | CCY 0327 | - | - | - | - | ++ | + |
| *Calothrix* sp. [a] | CCY 9923 | - | - | + | + | ++ | + |
| **Scytonemataceae cultures** | | | | | | | |
| *Scytonema hofmanni* [e] F | PCC 7110 | - | - | - | - | ++ | - |
| **Microchaetaceae cultures** | | | | | | | |
| *Microchaete* sp. [e,d,f] F | PCC 7126 | - | - | + | ++ | - | - |
| **Tolypothrichaceae cultures** | | | | | | | |
| *Tolypothrix tenuis* [e,d,f] F | PCC 7101 | - | - | ++ | + | - | - |

[revised manuscript text omitted]

Font: 10 pt

| Page 28: [2] Formatted | Jaap Damste | 11-11-2017 17:37:00 |
|---|---|---|

Line spacing:  Multiple 1.15 li

| Page 28: [3] Formatted | Jaap Damste | 11-11-2017 17:37:00 |
|---|---|---|

Font: 10 pt

| Page 28: [3] Formatted | Jaap Damste | 11-11-2017 17:37:00 |
|---|---|---|

Font: 10 pt

| Page 28: [3] Formatted | Jaap Damste | 11-11-2017 17:37:00 |
|---|---|---|

Font: 10 pt

| Page 28: [3] Formatted | Jaap Damste | 11-11-2017 17:37:00 |
|---|---|---|

Font: 10 pt

| Page 28: [3] Formatted | Jaap Damste | 11-11-2017 17:37:00 |
|---|---|---|

Font: 10 pt

| Page 28: [3] Formatted | Jaap Damste | 11-11-2017 17:37:00 |
|---|---|---|

Font: 10 pt

| Page 28: [3] Formatted | Jaap Damste | 11-11-2017 17:37:00 |
|---|---|---|

Font: 10 pt

| Page 28: [3] Formatted | Jaap Damste | 11-11-2017 17:37:00 |
|---|---|---|

Font: 10 pt

| Page 28: [3] Formatted | Jaap Damste | 11-11-2017 17:37:00 |
|---|---|---|

Font: 10 pt

| Page 28: [4] Formatted | Jaap Damste | 8-11-2017 10:35:00 |
|---|---|---|

English (U.S.)

| Page 28: [5] Formatted Table | Jaap Damste | 26-10-2017 22:33:00 |
|---|---|---|

Formatted Table

| Page 28: [6] Formatted | Jaap Damste | 8-11-2017 10:35:00 |
|---|---|---|

English (U.S.)

| Page 28: [7] Formatted | Jaap Damste | 8-11-2017 10:35:00 |
|---|---|---|

English (U.S.)

| Page 28: [7] Formatted | Jaap Damste | 8-11-2017 10:35:00 |
|---|---|---|

English (U.S.)

| Page 28: [8] Formatted | Jaap Damste | 8-11-2017 10:35:00 |
|---|---|---|

English (U.S.)

| Page 28: [8] Formatted | Jaap Damste | 8-11-2017 10:35:00 |
|---|---|---|

English (U.S.)

| Page 28: [9] Formatted | Jaap Damste | 8-11-2017 10:35:00 |
|---|---|---|

English (U.S.)

| Page 28: [10] Formatted | Jaap Damste | 8-11-2017 10:35:00 |
|---|---|---|

English (U.S.)

| Page 28: [11] Formatted | Jaap Damste | 8-11-2017 10:35:00 |
| --- | --- | --- |

English (U.S.)

| Page 28: [12] Formatted | Jaap Damste | 8-11-2017 10:35:00 |
| --- | --- | --- |

English (U.S.)

| Page 28: [13] Formatted | Jaap Damste | 8-11-2017 10:35:00 |
| --- | --- | --- |

English (U.S.)

| Page 28: [14] Formatted | Jaap Damste | 8-11-2017 10:35:00 |
| --- | --- | --- |

English (U.S.)

| Page 28: [15] Formatted | Jaap Damste | 8-11-2017 10:35:00 |
| --- | --- | --- |

English (U.S.)

| Page 28: [16] Formatted | Jaap Damste | 8-11-2017 10:35:00 |
| --- | --- | --- |

Underline, English (U.S.)

| Page 28: [16] Formatted | Jaap Damste | 8-11-2017 10:35:00 |
| --- | --- | --- |

Underline, English (U.S.)

| Page 28: [17] Formatted | Jaap Damste | 8-11-2017 10:35:00 |
| --- | --- | --- |

Underline, English (U.S.)

| Page 28: [18] Formatted | Jaap Damste | 8-11-2017 10:35:00 |
| --- | --- | --- |

English (U.S.)

| Page 28: [19] Formatted | Jaap Damste | 8-11-2017 10:35:00 |
| --- | --- | --- |

English (U.S.)

| Page 28: [20] Formatted | Jaap Damste | 8-11-2017 10:35:00 |
| --- | --- | --- |

English (U.S.)

| Page 28: [21] Formatted | Jaap Damste | 8-11-2017 10:35:00 |
| --- | --- | --- |

English (U.S.)

| Page 28: [22] Formatted | Jaap Damste | 8-11-2017 10:35:00 |
| --- | --- | --- |

English (U.S.)

| Page 28: [23] Formatted | Jaap Damste | 8-11-2017 10:35:00 |
| --- | --- | --- |

English (U.S.)

| Page 28: [24] Formatted Table | Jaap Damste | 26-10-2017 22:33:00 |
| --- | --- | --- |

Formatted Table

| Page 28: [25] Formatted | Jaap Damste | 8-11-2017 10:35:00 |
| --- | --- | --- |

English (U.S.)

| Page 28: [26] Formatted | Jaap Damste | 8-11-2017 10:35:00 |
| --- | --- | --- |

Underline, English (U.S.)

| Page 28: [26] Formatted | Jaap Damste | 8-11-2017 10:35:00 |
| --- | --- | --- |

Underline, English (U.S.)

| Page 28: [27] Formatted | Jaap Damste | 8-11-2017 10:35:00 |
| --- | --- | --- |

Underline, English (U.S.)

| Page 28: [28] Formatted | Jaap Damste | 8-11-2017 10:35:00 |
| --- | --- | --- |

English (U.S.)

| Page 28: [29] Formatted | Jaap Damste | 8-11-2017 10:35:00 |
| --- | --- | --- |

Underline, English (U.S.)

| Page 28: [29] Formatted | Jaap Damste | 8-11-2017 10:35:00 |

Underline, English (U.S.)

| Page 28: [29] Formatted | Jaap Damste | 8-11-2017 10:35:00 |

Underline, English (U.S.)

| Page 28: [30] Formatted | Jaap Damste | 8-11-2017 10:35:00 |

English (U.S.)

| Page 28: [31] Formatted | Jaap Damste | 8-11-2017 10:35:00 |

English (U.S.)

| Page 28: [32] Formatted Table | Jaap Damste | 26-10-2017 23:02:00 |

Formatted Table

| Page 28: [33] Formatted | Jaap Damste | 8-11-2017 10:35:00 |

English (U.S.)

| Page 28: [34] Formatted | Jaap Damste | 8-11-2017 10:35:00 |

English (U.S.)

| Page 28: [35] Formatted | Jaap Damste | 8-11-2017 10:35:00 |

English (U.S.)

| Page 28: [36] Formatted | Jaap Damste | 8-11-2017 10:35:00 |

English (U.S.)

| Page 28: [37] Formatted | Jaap Damste | 8-11-2017 10:35:00 |

English (U.S.)

| Page 28: [38] Formatted | Jaap Damste | 8-11-2017 10:35:00 |

English (U.S.)

| Page 28: [39] Formatted | Jaap Damste | 8-11-2017 10:35:00 |

Underline, English (U.S.)

| Page 28: [39] Formatted | Jaap Damste | 8-11-2017 10:35:00 |

Underline, English (U.S.)

| Page 28: [40] Formatted | Jaap Damste | 8-11-2017 10:35:00 |

Underline, English (U.S.)

| Page 28: [41] Formatted | Jaap Damste | 8-11-2017 10:35:00 |

English (U.S.)

| Page 28: [42] Formatted | Jaap Damste | 8-11-2017 10:35:00 |

English (U.S.)

| Page 28: [43] Formatted | Jaap Damste | 8-11-2017 10:35:00 |

Underline, English (U.S.)

| Page 28: [44] Formatted | Jaap Damste | 8-11-2017 10:35:00 |

English (U.S.)

| Page 28: [45] Formatted | Jaap Damste | 8-11-2017 10:35:00 |

English (U.S.)

| Page 28: [46] Formatted | Jaap Damste | 8-11-2017 10:35:00 |

Underline, English (U.S.)

| Page 28: [47] Formatted | Jaap Damste | 8-11-2017 10:35:00 |

English (U.S.)

| Page 28: [48] Formatted | Jaap Damste | 8-11-2017 10:35:00 |

English (U.S.)

| Page 28: [49] Formatted Table | Jaap Damste | 26-10-2017 22:33:00 |

Formatted Table

| Page 28: [50] Formatted | Jaap Damste | 8-11-2017 10:35:00 |

English (U.S.)

| Page 28: [51] Formatted | Jaap Damste | 8-11-2017 10:35:00 |

English (U.S.)

| Page 28: [52] Formatted | Jaap Damste | 8-11-2017 10:35:00 |

English (U.S.)

| Page 28: [53] Formatted | Jaap Damste | 8-11-2017 10:35:00 |

English (U.S.)

| Page 28: [54] Formatted | Jaap Damste | 8-11-2017 10:35:00 |

English (U.S.)

| Page 28: [55] Formatted | Jaap Damste | 8-11-2017 10:35:00 |

English (U.S.)

| Page 28: [56] Formatted | Jaap Damste | 8-11-2017 10:35:00 |

English (U.S.)

| Page 28: [57] Formatted | Jaap Damste | 8-11-2017 10:35:00 |

English (U.S.)

| Page 28: [58] Formatted | Jaap Damste | 8-11-2017 10:35:00 |

English (U.S.)